# Optimal density of bacterial cells

**Tin Yau Pang**[1,2], **Martin J. Lercher**[1] *

1 Institute for Computer Science & Department of Biology, Heinrich Heine University, Düsseldorf, Germany,
2 Division of Cardiology, Pulmonology and Vascular Medicine, Medical Faculty and University Hospital Düsseldorf, Heinrich Heine University, Düsseldorf, Germany

\* martin.lercher@hhu.de

**Data Availability Statement:** The source code used for simulations in this study are uploaded to two GltHub repositories, one for the linear model and the other for the whole cell model: 1 https://github.com/TinPang/optimalCytosolicDensity_

## Abstract

A substantial fraction of the bacterial cytosol is occupied by catalysts and their substrates. While a higher volume density of catalysts and substrates might boost biochemical fluxes, the resulting molecular crowding can slow down diffusion, perturb the reactions' Gibbs free energies, and reduce the catalytic efficiency of proteins. Due to these tradeoffs, dry mass density likely possesses an optimum that facilitates maximal cellular growth and that is interdependent on the cytosolic molecule size distribution. Here, we analyze the balanced growth of a model cell, accounting systematically for crowding effects on reaction kinetics. Its optimal cytosolic volume occupancy depends on the nutrient-dependent resource allocation into large ribosomal vs. small metabolic macromolecules, reflecting a tradeoff between the saturation of metabolic enzymes, favoring larger occupancies with higher encounter rates, and the inhibition of the ribosomes, favoring lower occupancies with unhindered diffusion of tRNAs. Our predictions across growth rates are quantitatively consistent with the experimentally observed reduction in volume occupancy on rich media compared to minimal media in *E. coli*. Strong deviations from optimal cytosolic occupancy only lead to minute reductions in growth rate, which are nevertheless evolutionarily relevant due to large bacterial population sizes. In sum, cytosolic density variation in bacterial cells appears to be consistent with an optimality principle of cellular efficiency.

## Author summary

The cellular cytosol harbours diverse molecules, whose crowding slows down diffusion and perturbs the chemical equilibrium of biochemical reactions. Reaction rates thus depend not only on the reactants themselves, but also on the background density of other molecules; as a consequence, maximal cell growth requires an optimal density. Here, we simulate a model cell with crowding-adjusted metabolic reaction kinetics. Its cytosol accommodates two types of reactions: metabolic reactions, involving small molecules, and protein production reactions that involve much larger molecules. These two cellular subsystems have distinct optimal densities, and a shift in their relative contribution to the cellular biomass explains the observed 10% difference in cytosolic density between *E. coli* bacteria growing in nutrient-rich and -poor environments.

linearModel 2 https://github.com/TinPang/
optimalCytosolicDensity_wholeCellModel.

**Funding:** This work was supported by DFG Grants
CRC 1310 to T.Y.P. and M.J.L. and by a grant of
the Volkswagen Foundation in the "Life?" initiative
to M.J.L.. This work was also supported, in part, by
the MODS project funded from the programme
"Profilbildung 2020" (grant number
PROFILNRW2020-107-A), an initiative of the
Ministry of Culture and Science of the State of
North Rhine-Westphalia awarded to M.J.L.. The
funders had no role in study design, data collection
and analysis, decision to publish, or preparation of
the manuscript. The grant from Volkswagen
Foundation supported the salary of TYP.

**Competing interests:** The authors have declared
that no competing interests exist.

## Introduction

The dry mass dissolved in the major compartment of bacterial cells, the cytosol, comprises hundreds of molecular species, including proteins, metabolites, polysaccharides, and nucleic acids. These molecules can be roughly classified into two sectors: the ribosomal sector, dominated by ribosomes and tRNA; and the non-ribosomal sector, comprising mostly metabolites, enzymes, and other proteins [1]. The molecules in these two sectors have very different size distributions: the ribosome is 65 times larger than the median enzyme size (2,600kDa [2] vs. 40kDa [3]), and tRNAs are about 300 times larger than typical metabolites (26kDa [4] vs. 89Da, the mass of alanine). The allocation of dry mass between the two sectors of the cellular economy can be summarized by a single parameter, the growth rate $\mu$, with the dry mass fraction of the ribosomal sector increasing almost linearly with $\mu$ [1,5]. Accordingly, the ribosome-rich cytosol at fast growth in nutrient-rich environments and the ribosome-meager cytosol at slow growth in nutritionally poor environments exhibit very different distributions of molecule sizes.

There are multiple approaches to measure the cytosolic dry mass density. Experiments based on centrifugation directly measured the buoyant density of cells, reporting that the *E. coli* cell density is approximately independent of growth rate across different nutritional conditions [6,7]. Experiments based on optical intensity measurements of cellular dry weight and cellular volume of *E. coli* also show an approximate proportionality between weight and volume, again suggesting a single dry mass density across conditions [8]. Recently, more accurate experiments that employed spatial light interference microscopy [9] confirmed an approximately constant dry mass density at 300g/L across minimal nutrient conditions with low growth rates. However, the same study found a roughly 10% less dense cytosol in a rich medium supporting high growth rates [10].

How does the cell control its density during cellular growth? Analyses of individual cells showed that both the cell width and the surface area / mass ratio of a cell are approximately constant throughout its growth trajectory and depend on the nutritional environment [10]. When the nutrients change, the cell changes its molecular machinery to adapt. These changes lead to a shift in Turgor pressure in the cell and change its width. At the same time, the cell also actively shifts the coupling between cell-wall insertion and biomolecule synthesis, leading to a different surface / mass ratio. Essentially, these mechanisms allow the cell to keep a roughly constant density over generational time scales, despite short-term density variations along the growth trajectory. In the current analysis, we are not concerned with the regulatory details of this density homeostasis, but we want to explore how natural selection might influence the target density of the cellular regulation.

We hypothesized that the difference in cytosolic density between slow and fast growth observed by Oldewurtel et al. [10] may be an evolutionary consequence of the differences in molecular composition. Cellular physiology has evolved under natural selection; thus, if the level of molecular crowding—the dry mass density of the cytosol—affects cellular efficiency and hence fitness, we expect density regulation to have evolved to a near-optimal, possibly condition-dependent state. While previous work has explored the influence of (macro-)molecular crowding on bacterial physiology and growth rates, these analyses assumed a constant, hard limit on the total cytosolic protein [11,12] or dry mass [13,14] concentration. These works do not justify the existence and magnitude of the density limit based on physicochemistry, and they cannot explain differences in the level of crowding (dry mass density) across conditions.

Biochemical reaction fluxes typically increase with increasing encounter rates of the molecule species involved; ignoring crowding effects on diffusion, encounter rates increase with

increasing density of the respective molecules. At the same time, the molecular crowding caused by other molecule species in the background (volume-excluding co-solutes) affects fluxes in at least three distinct ways. (i) Crowding slows down the diffusion of a catalyst and its substrates, thereby reducing their encounter rates [15,16]. (ii) Crowding limits the available volume and thus reduces a solute's entropy (the "excluded volume phenomenon"), thereby changing the free energies of the molecules involved in the reaction and consequently shifting the equilibrium concentrations of substrates and products [17]. (iii) Crowding can affect the structure of the protein catalyst, the process of its folding, and its conformational stability [18]; these structural changes may disturb the reaction flux if they affect the active site [18–22]. Due to these opposing effects, there may be an optimal cytosolic density where cellular efficiency and hence fitness are maximal.

The effect of crowding depends on the size of the catalyst and its substrate: in the presence of other volume-excluding cosolutes, the larger the size of a solute, the stronger the reduction of its diffusion [23] and the perturbation of its free energy [24,25]. Hence, when the size distribution of the molecules changes, the cell needs to adjust its cytosolic density in order to optimize its physiological efficiency.

In a pioneering theoretical study, Vazquez [26] considered how the objective flux of a metabolic network is affected by crowding, assuming that the enzymes are also the crowders in their own right. The corresponding model accounts for the slowdown of diffusion due to crowding, but ignores the growth-rate dependent role of the ribosomal sector and the corresponding changes in the molecular size distribution. This study found that there exists an optimal cytosolic density, which maximizes reaction fluxes and depends on details of the network, such as the ratio between diffusion limited and transition-state limited reactions. Using an alternative modeling approach, Dill et al. also arrives at a similar conclusion [27].

While these studies demonstrated the existence of a flux-optimizing cytosolic density, natural selection maximizes fitness, not metabolic reaction fluxes. For non-interacting cells in a uniform environment, fitness is closely related to the growth rate [28]. The cellular physiology at maximal growth rate—and hence maximal fitness—can be described mathematically through growth balance analysis (GBA) [13].This modeling framework simulates the balanced growth of a self-replicating bacterial cell while accounting for the major physicochemical constraints on cellular growth: mass balance of metabolism and protein production, non-linear reaction kinetics that depend on the concentrations of catalysts and their substrates, and the effects of molecular crowding. Vazquez [26] assumed that all crowding effects can be quantified by classifying reactions into two types: (1) those in the saturation regime, with $[S] \gg K_M$, and (2) those in the diffusion limited regime, with $[S] \ll K_M$. This approach constrains the modeled substrate concentrations of each reaction to be either much smaller or much larger than the corresponding $K_M$, which is incompatible with a realistic modeling of intracellular metabolite concentrations and their effect on molecular crowding [29].

Standard GBA assumes a given, hard limit on dry mass density [13,14] or on total protein concentration [12]. Here, to assess the effects of molecular crowding on fitness, we apply a generalization of GBA that instead describes the kinetics of metabolic reactions and protein translation through crowding-adjusted Michaelis-Menten kinetics [25,30]. We maximize the balanced growth rate while varying the concentrations of transporters, catalytic proteins, and metabolites; these molecular species also form the volume excluding co-solutes in the background of each reaction, affecting diffusion, free energies, and hence reaction kinetics through molecular crowding. Consistent with experimental observations, we find that the cytosolic density of optimal growth strategies depends on the cellular growth rate.

## Results

### Crowding-adjusted reaction kinetics

The effects of molecular crowding on biochemical reaction kinetics are due to volume exclusion effects. Thus, the relevant parameter for their quantification is not the total mass density (dry mass per volume) but the total volume occupancy $\rho$, i.e., the fraction of cytosolic volume occupied by dry mass. However, as molecular mass and volume are approximately proportional [31], we treat density and occupancy as interchangeable, subject to a scaling coefficient that quantifies the mass/volume ratio of the cytosolic dry mass. Depending on the external conditions, the volume occupancy of biopolymers in the cytosol of *E. coli* lies within a range 0.16–0.36 [32], providing a lower limit of the total volume occupancy.

To model biochemical reaction kinetics as a function of molecular crowding, we use a modified description of irreversible Michaelis-Menten kinetics [25,30] (see Methods for details),

$$v = k_{\text{cat}} \frac{[s][E]}{K_{\text{M}}^* + [s]},\tag{1}$$

where $[S]$ and $[E]$ are the concentrations of the substrate and catalyst, respectively; $k_{\text{cat}}$ is the catalytic rate constant (turnover number); and $K_{\text{M}}^*$ is the crowding-adjusted Michaelis parameter [25, 30]. In a typical catalytic reaction, a substrate molecule $S$ encounters a catalyst molecule $E$ to form a catalyst-substrate complex $ES$, which either proceeds forward to convert the substrate into the product, or reverts back to release the substrate. To derive $K_{\text{M}}^*$, we assume that a catalytic reaction can be divided into two subsequent, independent steps that both depend on the cytosolic volume occupancy: (1) $S$ and $E$ diffuse until their encounter (Fig 1B) and thereafter (2) they bind and unbind reversibly until the reaction proceeds forward and product $P$ is released (Fig 1A). In step (2), we further assume that $S$ and $E$ will stay in close proximity and do not diffuse away from each other. These approximations simplify the model derivation and make the reaction times of the two steps additive [25]. Combining the rate laws of the two steps provides an estimate for the effective Michaelis parameter $K_{\text{M}}^*$ [25,30]:

$$K_{\text{M}}^* = K_{\text{M}}^0 \frac{\Gamma + \theta \exp(-g\rho)}{(1 + \theta)\Gamma \exp(-g\rho)}\tag{2}$$

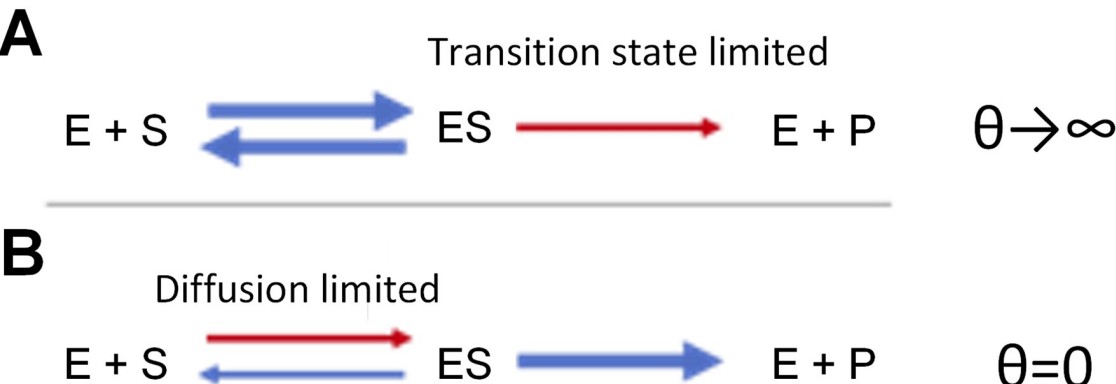

**Fig 1. Limiting cases of crowding effects on kinetics. (A)** Transition state limit, valid if the conversion of enzyme (E) plus substrate (S) to the complex ES is much faster than the conversion of ES to product (P). This limit corresponds to $\theta \to \infty$ in the whole cell model. **(B)** Diffusion limit, where the conversion of ES to P is much faster than the formation of ES. It corresponds to $\theta = 0$ in the whole cell model.

Here, $K_M^0$ is the Michaelis parameter in the low-crowding limit; $\Gamma$ is the correction term for the shift in Gibbs free energy $\Delta G$ due to molecular crowding, with $\Gamma = 1$ in the limit of low molecular crowding; the exponential term $\exp(-g\rho)$ accounts for the slow-down of diffusion, where the scaling factor $g$ can be estimated from the size of the substrate, i.e., $g = g(r_S)$ for a spherical substrate with radius $r_S$; and $\theta$ is the relative weight between the rate laws of steps (1) and (2), or in other words the relative ratio between time spent on step (1) and on step (2) at low cytosolic occupancy. Transition state limitation dominates when $\theta$ is large ($\theta \to \infty$; Fig 1A), this gives $K_M^* \simeq K_M^0 \Gamma$; diffusion limitation dominates when $\theta$ is small ($\theta \to 0$; Fig 1B), this gives $K_M^* \simeq K_M^0 \exp(-g\rho)$. The best available estimate for $\theta$ is 2.3, obtained for the ERK MAP kinase phosphorylation reaction [30]; we employ this value in our simulations for metabolic and ribosomal reactions. Small metabolites, however, may diffuse more readily, and so metabolic reactions may have a stronger bias towards being transition state-dominated than ribosomal reactions. Therefore, we also employ an alternative model, in which the metabolic reaction has a $\theta$ twice as large as that of ribosomal reactions ($\theta = 4.6$ for metabolic and $\theta = 2.3$ for ribosomal reactions), and examine if this alternative assumption affects our conclusions.

## Protein translation favors lower occupancy compared to metabolic pathways

To understand the effect of crowding on catalytic reactions, we first consider a simple, linear biochemical pathway model consisting of $N = 20$ consecutive enzyme-catalyzed reactions at steady state, i.e., with identical flux through each reaction (Fig 2A). The kinetics of each step in the pathway depend not only on the concentrations of the catalyzing enzyme and the substrate of the reaction, but—through the crowding effects on $K_M^*$—also on the concentrations of the molecules involved in the remaining $N$-1 reactions. We identified the combination of enzyme and metabolite concentrations that maximizes the pathway output per dry mass, calculated from crowding-adjusted kinetics; adding these concentrations, weighted by the respective molecular volumes, resulted in an estimate of the optimal cytosolic occupancy $\rho_{opt}$ for this system. To restrict the number of model parameters, we assumed identical enzyme and substrate sizes as well as identical crowding-adjusted kinetics (Eq (1)) for all reactions in the pathway. We fixed the physicochemical parameter $K_M^0 = 130\mu M$, which is the median value for metabolic enzymes and their substrates [33] and is also close to the value estimated for the binding of ternary complexes to the ribosome [34], 120μM. The second physicochemical parameter in this model, $k_{cat}$, appears merely as a scaling factor of the pathway flux, and for simplicity we set $k_{cat} = 1s^{-1}$ without loss of generality.

We varied the sizes of the catalyst and its substrate, which we assumed to be both spherical. We considered (i) a metabolic pathway, where substrates have sizes typical for metabolites ($r = 0.34nm$) and catalyst have sizes typical for globular proteins ($r = 2.4nm$); and (ii) a ribosomal system with sizes resembling those of tRNAs ($r = 2.4nm$) and ribosomes ($r = 13nm$). We also assumed the catalyst-substrate complex to be spherical, with a volume equal to that of the catalyst plus the substrate. Note that while cells do not contain pathways of consecutive ribosomes, the model for the 20-step linear pathway depicted in Fig 2A is mathematically identical to the model for 20 parallel reactions shown in Fig 2B if all metabolite concentrations are assumed to be identical; this latter model can be interpreted as a molecular snapshot showing 20 types of ribosomes, each presenting an anticodon for a different tRNA species.

Fig 3A plots the reaction fluxes. In the metabolic system (blue), the total reaction flux increases monotonically with the cytosolic occupancy within the range explored; note that the term that describes the approximate effects of diffusion, $\exp(-g\rho)$, breaks down close to $\rho = 1$ [16,30]. In contrast, in the ribosomal system (red), the total reaction flux reaches a maximum

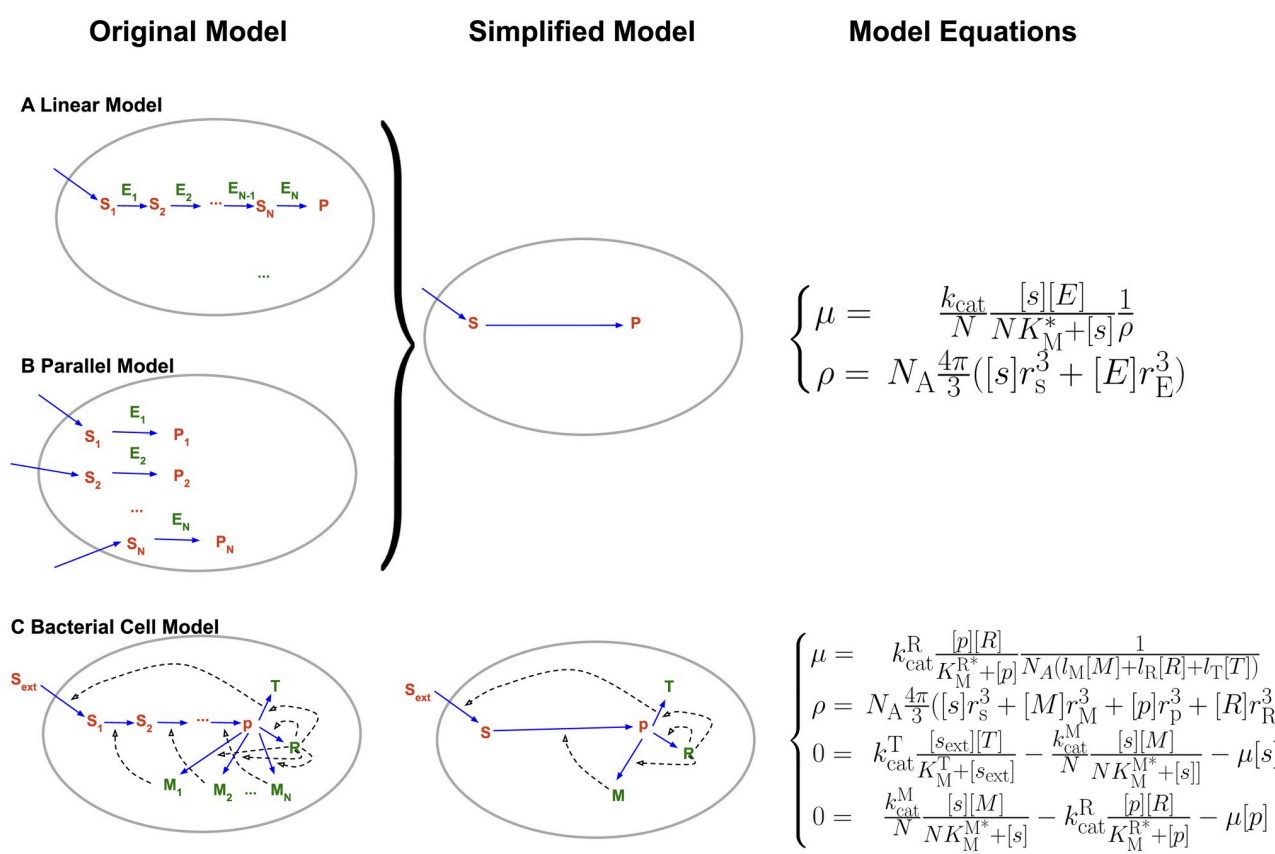

**Fig 2. Illustration of the models. (A) Reaction system in an *N*-steps linear pathway**, representing a metabolic system. The initial substrate, $s_1$, is replenished by a transport process not included in the model, and is converted by the pathway into the product $p$. **(B) Reaction system of *N* parallel reactions**, representing ribosomes presenting *N* distinct anticodons; $s_1, \ldots, s_N$ are the corresponding ternary complexes, $p_1, \ldots, p_N$ are the extending amino acid chains. Assuming that (i) all reactions follow identical kinetics, (ii) all substrate concentrations $s_i$ are identical, and (iii) all enzyme concentrations are identical, the fluxes of the models in (A) and (B) are both mathematically identical to the simplified model to the right of the two panels, defined in Eqs (S7) and (S8), with re-scaled concentrations $[s] = [s_1]+[s_2]+\ldots+[s_N]$ and $[E] = [E_1]+[E_2]+\ldots+[E_N]$. **(C) GBA model simulating the balanced growth of a bacterial cell**. Transporter *T* imports nutrient $s_1$, which is converted to the precursor for protein production $p$ by a metabolic pathway with *N* consecutive enzymes, $M_1, \ldots, M_N$, via intermediate substrates $s_2, \ldots, s_N$. The ribosome *R* synthesizes the *N*+2 proteins (*T*, *R*, $M_1, \ldots, M_N$) from $p$. Assuming identical concentrations of metabolic enzymes, $[M_1] = \ldots [M_N] =: [M]$, and of metabolites, $[s_1] = \ldots [s_N] =: [s]$, the solution space of this model cell is spanned by the five concentrations $[s],[p],[T],[M],[R]$ and satisfies Eqs (S9), (S10) and (S11). Here $l_T$, $l_M$, and $l_R$ are the number of precursor molecules required to synthesize a transporter protein, a metabolic enzyme, and a ribosome in the model, with values 300, 300, and 7.459, respectively.

at $\rho = 0.21$, beyond which potential flux increases due to more and more highly saturated ribosomes are drowned out by increasingly difficult diffusion. Fig 3B plots the flux per biomass investment, a proxy for growth rate, defined as the reaction fluxes divided by cytosolic occupancy. Here, both systems show a clear optimum, with an optimal occupancy of $\rho_{opt} = 0.30$ for the metabolic system and $\rho_{opt} = 0.12$ for the ribosomal system. In the light of natural selection on the cellular growth rate, optimal growth–which occurs at a different occupancy than maximal flux–was likely more relevant.

The effect of the occupancy on the pathway output is substantial in the ribosomal system, where a 50% drop in occupancy from $\rho_{opt}$ decreases pathway flux per dry mass by 21%, while it is small for the metabolic system, where a 50% drop in occupancy incurs a flux decrease per dry mass of only 1.3%. Increasing the number of steps in the pathway from 20 to 100 has almost no effect on the optimal occupancies $\rho_{opt}$ for both systems; however, the reduction of flux per dry mass when $\rho$ deviates from $\rho_{opt}$ is substantially increased for longer pathways or

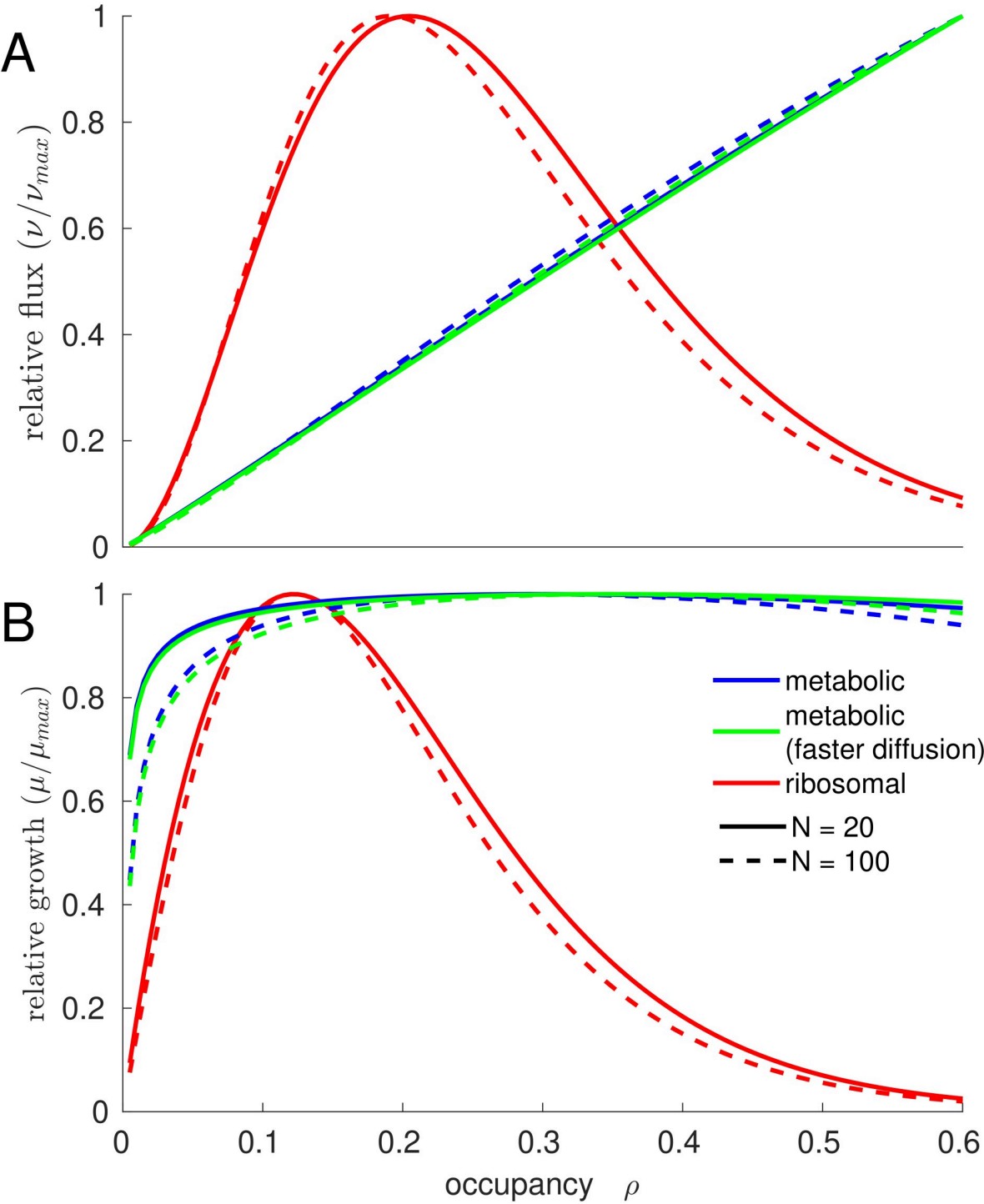

**Fig 3. The cytosolic occupancy that facilitates maximal biochemical reaction fluxes is lower for ribosomal than for metabolic systems.**
(A) reaction fluxes and (B) growth rate (flux per unit dry mass) of the linear model. Blue and green lines represent metabolic systems, with small catalysts and substrates, where blue is for $\theta$ = 2.3 and green is for $\theta$ = 4.6 (stronger bias towards transition state limitation due to fast diffusion). Red lines represent ribosomal systems with much larger molecules. Solid lines are for systems of $N$ = 20 consecutive (metabolic, blue) or parallel (ribosomal, red) reactions, dashed lines are for larger systems of $N$ = 100 reactions.

more parallel reactions (Fig 3, dashed lines). Assuming that metabolic reactions are more biased towards being transition state limited (green curves in Fig 3) has only very small effects. Moreover, the approach to crowding employed in Vazquez [26] also leads to conclusions that are qualitatively similar to Fig 3 (S1 Fig). We conclude that the model predictions regarding the differences between metabolic and ribosomal systems are robust and do not depend on model details.

We conclude that the results from the pathway model are consistent with the experimentally observed trend that the cytosol of fast-growing bacteria, which is dominated by the larger molecules of the ribosomal sector, favors a lower occupancy than the cytosol of a slowly growing cell dominated by smaller metabolites and enzymes. Moreover, our results suggest that compared to cells dominated by the ribosomal sector, cells dominated by the smaller molecules of the metabolic sector may suffer a smaller reduction in biochemical efficiency when the cytosol shifts away from $\rho_{opt}$.

## Optimal occupancy for a self-replicating cell at balanced growth

Can the single-pathway results be generalized to more realistic models of cellular growth, which combine metabolic and ribosomal activities, and where we can directly assess the effect of concentration changes on cellular growth rates? How does the cytosolic occupancy optimal for growth, $\rho_{opt}$, change when the number of active metabolic reactions changes, e.g., when switching from a minimal medium, where all biomass components have to be synthesized from a single carbon source through multi-step biochemical pathways, to a rich medium that provides many cellular building blocks through simple transport processes?

To answer these questions, we considered a schematic GBA model of a bacterial cell [13], the cytosol of which comprises interdependent ribosomal and metabolic sectors (Fig 2C). In this model cell, a nutrient $s_{ext}$ is imported into the cell by transport protein $T$; the nutrient is then converted into precursor $p$ by an $N$-steps metabolic pathway; finally, the ribosome $R$ uses $p$ to synthesize all catalytic proteins, including $T$, $R$, and the metabolic enzymes $M_1$ to $M_N$. For simplicity, all molecules are again assumed to be spherical in shape. All macromolecules, except for the transporter protein $T$, are located in the cytosol and are thus crowders in their own right; $T$ is assumed to be fully integrated into the membrane and does not contribute to crowding. As before, the molecules of the metabolic sector are small (metabolites $s_i$ with $r_s = 0.34$nm and metabolic enzymes $M_i$ with $r_M = 2.4$nm), while the constituents of the ribosomal sector are much larger (precursor $p$ with $r_p = 2.4$nm and ribosome $R$ with $r_R = 13$nm).

To estimate the appropriate number of active enzymatic reactions in the metabolic sector, $N$, we used flux balance analysis constrained by enzyme concentration [35]; simulations were performed with an improved implementation parameterized for a genome-scale model of *Escherichia coli* metabolism [36]. We found 259 active metabolic enzymes for growth in a minimal medium with glucose as the sole carbon source; 206 active enzymes for the same medium supplemented with amino acids; and 174 active enzymes for growth in a rich medium (Methods).

To facilitate the numerical determination of the state with maximal growth rate, we approximate the metabolic pathway through a single, lumped reaction with catalyst $M$ and substrate $s$, scaling kinetic parameters and molecular masses to account for the pathway length $N$; this approximation neglects the dilution of intermediate metabolite concentrations through volume growth (Methods). The solution space of the model with $N$ metabolic enzyme-catalyzed reactions is spanned by 5 concentration variables (Fig 2C), describing the transporter protein with concentration $[T]$, the cytosolic substrates with identical concentrations $[s_1] = \ldots = [s_N] =: [s]$, the protein precursor ("ternary complex") with concentration $[p]$, the metabolic

enzymes with identical concentrations $[M_1] = \ldots = [M_N] =: [M]$, and the ribosome with concentration $[R]$.

The metabolic and ribosomal reactions are described by crowding-adjusted irreversible Michaelis Menten kinetics (Eq (1)), whereas the transporter reaction is described by conventional irreversible Michaelis-Menten kinetics with constant $K_M^T$. We investigated how the cytosolic occupancy that allows the fastest growth varies with two parameters, $N$ and $[s_{ext}]$. $N$ is the number of enzyme species in the metabolic pathway, and is inversely related to the richness of the nutrient composition and entering the model through scaling the lumped metabolic reaction. $[s_{ext}]$ is the concentration of the external nutrient, which parameterizes the degree to which the available nutrients are limiting growth. As above, we challenged the assumptions of our model by performing a second set of simulations, allowing for a stronger bias of metabolic reactions towards transition state limitation ($\theta = 4.6$) compared to ribosomal reaction ($\theta = 2.3$).

## Optimal occupancy is lower at faster growth due to higher protein translation demands

In the whole-cell model, the optimal occupancy $\rho_{opt}$ generally falls between 0.1 and 0.3 (Fig 4). $\rho_{opt}$ increases when the number of simultaneously active metabolic reactions $N$ increases or when the external nutrient concentration $s_{ext}$ (and consequently the growth rate) decreases (Fig 5). These effects are due to shifts in the relative dry mass fractions of the metabolic sector (metabolic enzyme $M$ and the substrate $s$; favoring higher occupancy) and the ribosomal sector (ribosome $R$ and the precursor $p$; favoring a relatively lower occupancy). Using different $\theta$ values for the metabolic and ribosomal systems does not change the qualitative trends of the model (S2 Fig). At increasing $N$ and constant $s_{ext}$, the fraction of the cytosolic volume occupied by the metabolic sector expands at the expense of the ribosomal sector (S3A, S3B and S4A Figs), and the saturation of the ribosome with its substrate drops, whereas the saturation of the metabolic enzymes with their substrates varies within a small range (S4B Fig). At constant $N = 250$ and improving nutrient conditions (increasing $s_{ext}$), the fraction of the cytosol occupied by the metabolic sector also expands slightly at the expense of the ribosomal sector (S5A Fig), while the saturation of both metabolic enzymes and ribosomes increases (S5B Fig). Note that while the cytosolic concentration of ribosomes decreases with increasing growth rate (S6B Fig), the ribosomal proteome fraction increases (S6C Fig), as the model cell re-allocates resources from metabolic enzymes and transporters into ribosomes, mirroring experimental observations [1,37].

At optimal occupancy $\rho_{opt}$ for different $N$ and $s_{ext}$, the crowding-adjusted Michaelis parameter $K_M^*$ of the ribosomal reaction shows a marked increase with $\rho_{opt}$ (Fig 6; two-sided Spearman rank correlation coefficient $r = 0.995$, $P < 10^{-15}$), consistent with our observations of a strong dependence of the ribosomal flux on $\rho$ in the simple pathway model (Fig 3). In contrast, the $K_M^*$ of the metabolic reactions is almost invariant when plotted against $\rho_{opt}$ ($r = 0.034$, $P = 0.85$). These observations support the intuitive notion that the optimal occupancy in the whole-cell model reflects a tradeoff between the saturation of metabolic enzymes, favoring larger occupancies with higher encounter rates, and the inhibition of the ribosomes, favoring lower occupancies with unhindered diffusion of tRNAs. In agreement with our findings for the simple pathway model, the dependence of the growth rate on $\rho$ appears to be moderate: e.g., at $N = 150$ and $s_{ext} = 1\mu M$, a 50% reduction of $\rho$ from $\rho_{opt}$ reduces $\mu$ by only 1.1% (Fig 4).

The slow-down of diffusion and the perturbation of Gibbs free energies have opposing effects on reaction efficiencies. To consider these two effects separately, we define the transition state-perturbation only Michaelis parameter as given by Eq (2) when setting the diffusion scaling exponent to $g = 0$, and we define the diffusion-perturbation only Michaelis parameter

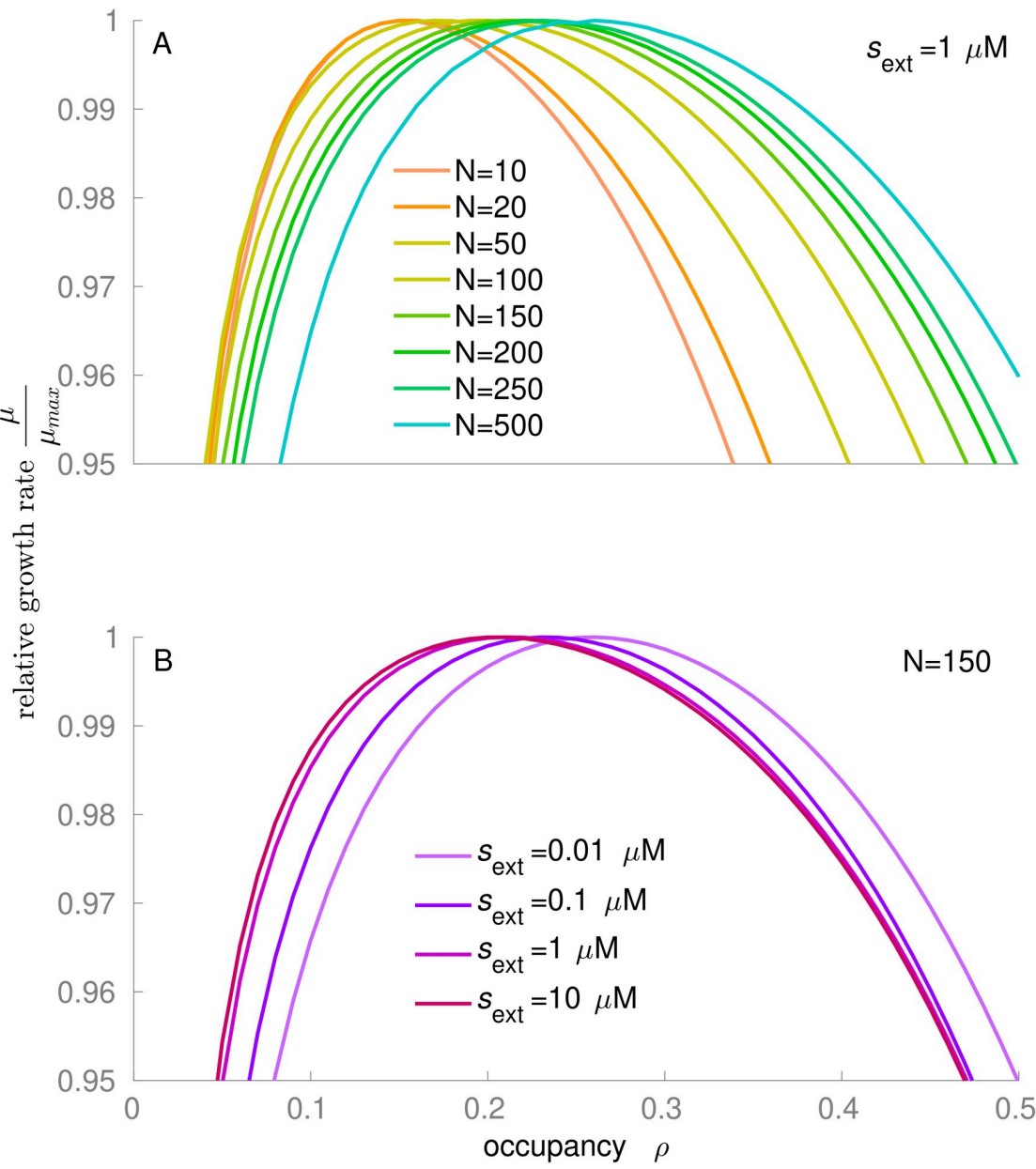

**Fig 4. The growth rate $\mu$ of the crowding-adjusted whole-cell model constrained at different cytosolic occupancies $\rho$.** The optimal cytosolic occupancy $\rho_{\text{opt}}$ **(A)** increases with $N$ (the number of enzymes in the metabolic pathway) and **(B)** decreases with external nutrient concentration $[s_{\text{ext}}]$. $\mu_{\text{max}}$ is the maximal growth rate for each curve across occupancies.

as given by Eq (2) when setting the Gibbs perturbation term to $\Gamma = 1$. We used these hypothetical Michaelis parameters in renewed simulations of the pathway models of the metabolic and ribosomal systems (Fig 2A and 2B). In the diffusion-perturbation only model, crowding increases $K_{\text{M}}^*$ and reduces the reaction fluxes (S7 Fig, dotted lines); in contrast, crowding in the transition state-perturbation only model decreases $K_{\text{M}}^*$ and boosts the fluxes (S7 Fig, dashed lines). At high occupancies ($\rho \gtrsim 0.5$), the transition state-perturbation effects reach a plateau, whereas the diffusion-perturbation effects continue to increase. Thus, at larger occupancies, the flux-reducing slow-down of diffusion always dominates over the flux-enhancing

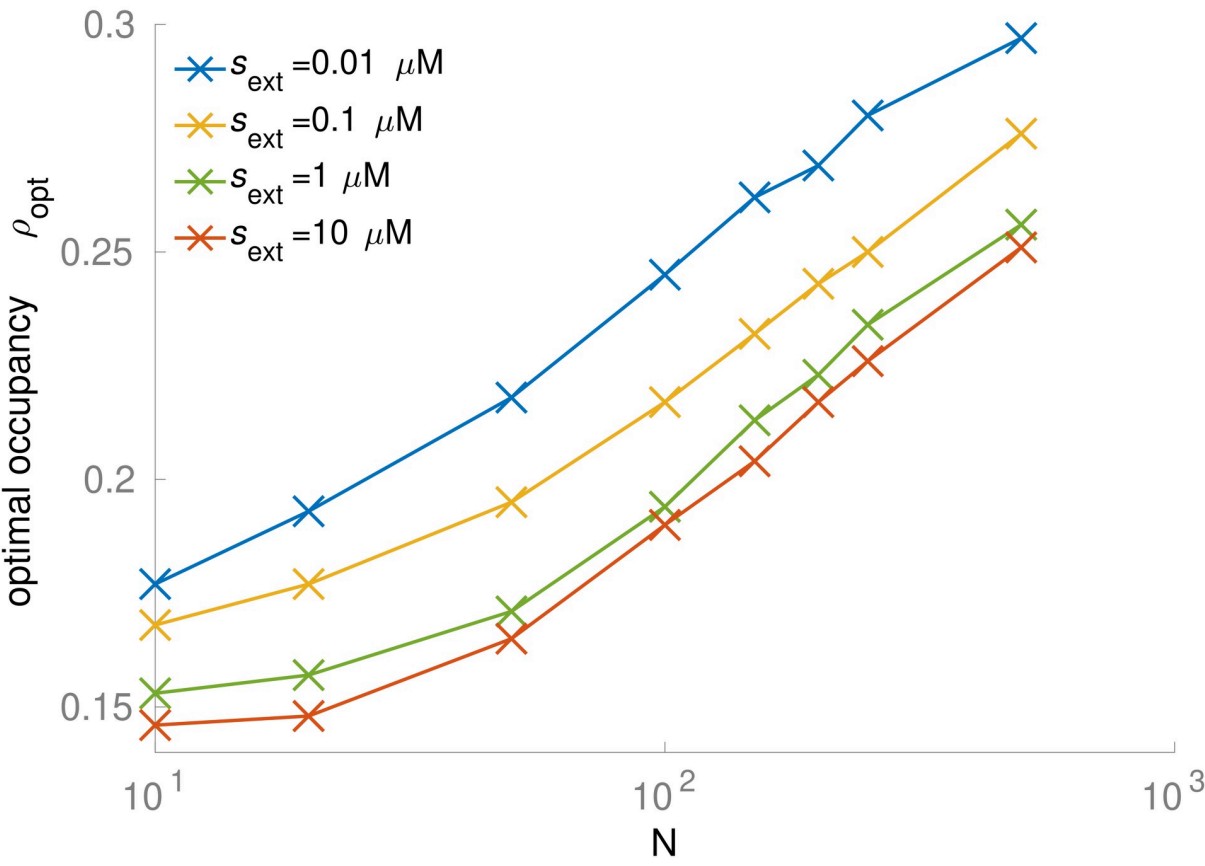

**Fig 5. The optimal cytosolic occupancy increases with metabolic pathway length $N$ and decreasing external nutrient concentration $s_{ext}$.**
The simulation is performed based on the whole cell model, in which both the metabolic reactions and ribosomal reactions have the same bias between diffusion limitation and transition state limitation. See S2 Fig for the simulation of a modified model, in which the metabolic reactions are more diffusion efficient.

perturbation of Gibbs free energy when considering their joint effect (S7 Fig, solid lines). In addition, shifting the bias toward stronger transition state perturbation (from solid blue to solid green) delays but does not change the overall trend of $K_M^*$ increases with increasing occupancy ρ. Comparison of S7A and S7B Fig shows that these trends are largely independent of $N$, the number of reactions in the system. While the trend for the slow-down of diffusion depends on the sizes of the reacting molecules, the trend for the perturbation of Gibbs free energies is largely independent of molecule sizes.

## Optimal occupancy is quantitatively consistent with the observed E. coli dry mass density

To compare our predictions of optimal occupancy to experimental data from *E. coli*, we first consider the transition from slow to fast growth on minimal media (simulated extracellular nutrient concentration $s_{ext} = 0.1 \mu M$ vs. $1 \mu M$, at a constant number of active metabolic reactions $N = 250$). Here, the optimal cytosolic volume occupancy $\rho_{opt}$ predicted by our whole-cell model decreases from 0.250 down to 0.234 (a 7% drop, Fig 5).

While no direct experimental observations of the occupancy ρ are available, occupancy is expected to scale approximately in proportion to the cytosolic dry mass density, $\rho_{DM}$ [31]. Oldewurtel et al. traces the $\rho_{DM}$ along the growth trajectories of wildtype *E. coli* cells (MG1655)

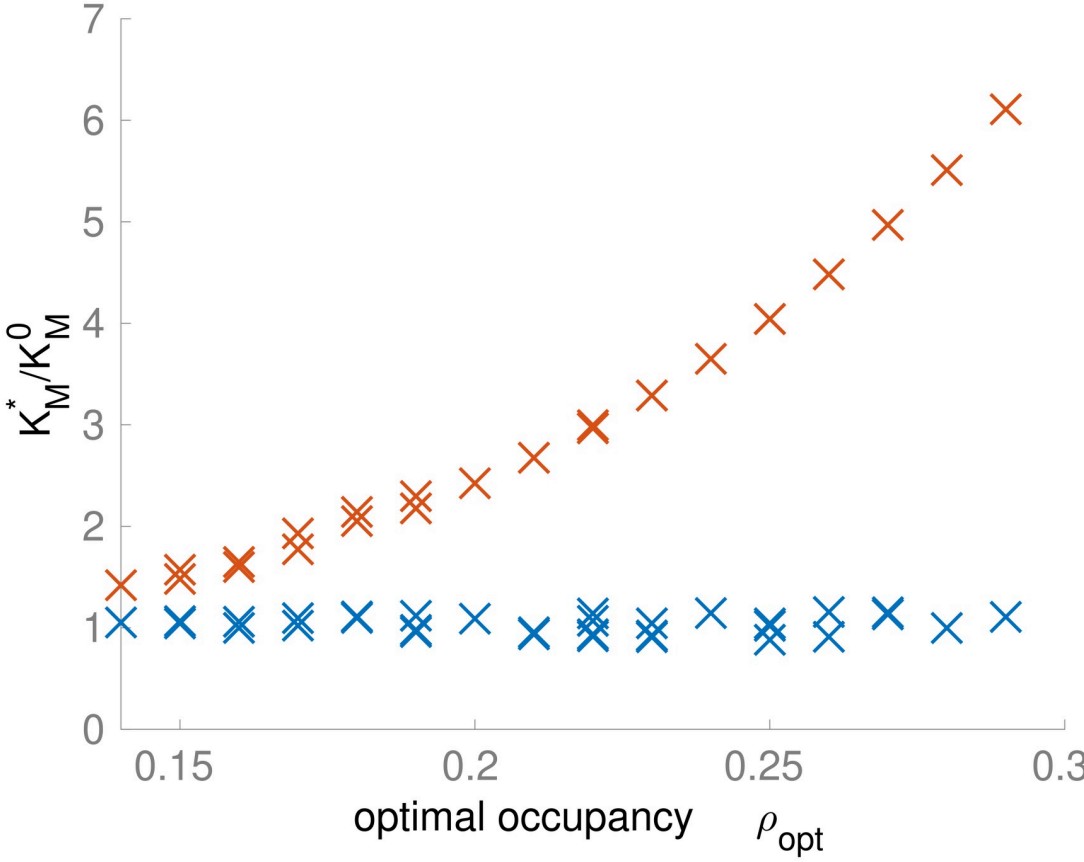

**Fig 6. The optimal occupancy strongly influences the effective Michaelis parameter $K_M^*$ of the ribosomal (red) but not of the metabolic (blue) reactions in the whole-cell model.** Each data point corresponds to a different combination of external nutrient concentration $s_{ext}$ and number of active metabolic reactions $N$. While the $K_M^*$ of the metabolic reactions does not correlate with $\rho_{opt}$ (blue markers; two-sided Spearman's rank correlation coefficient $r = 0.034$, $P = 0.85$), the $K_M^*$ of the ribosomal reactions correlates with $\rho_{opt}$ (red markers; $r = 0.998$, $P<10^{-15}$). The discrete distribution of points along the $x$-axis reflects the step size used for $\rho$ in the simulations.

cultured in different minimal and in rich medium (see Fig 2B of Oldewurtel et al. [10]). S8A Fig summarizes the mass density distributions for different conditions, and S1A Table shows the results of Wilcoxcon rank sum tests that compare the median mass densities across conditions. The measurements indicate that $\rho_{DM}$ does not vary noticeably with $\mu$ across most minimal media, i.e., $\rho_{DM}\approx31$g/mL when $\mu\leq0.7$h$^{-1}$. While the mass density in growth on minimal medium is statistically significantly different from that in the other two minimal media conditions, the observed $\rho_{DM}$ distributions are strongly overlapping (S8A Fig). In contrast, the observed distribution of mass density drops to a much lower mean value in rich medium, $\rho_{DM}$ = 0.28g/mL, where $\mu = 1.2$h$^{-1}$. To compare this data to our predictions, we have to convert mass densities to occupancy values (volume density). As cellular resources are shifted from protein to ribosomal RNA and tRNA with increasing growth rate, and because RNA is denser than protein, the conversion factor between dry mass and volume changes with growth rate (Methods). Taking this into account, we find that the experimental data indicates a decrease of $\rho$ from 0.223 to 0.215 (S1B and S8B Figs), a 4% reduction that is consistent with our prediction of 7% (Fig 5).

Second, we consider the transition from fast growth on minimal media to ultrafast growth in rich medium (simulated extracellular nutrient concentration $s_{ext}$ of 1μM vs. 10μM, number of active metabolic reactions $N = 250$ vs. $N = 150$). Here, the predicted optimal cytosolic volume occupancy $\rho_{opt}$ decreases by 15%, from 0.234 to 0.204 (Fig 5). Empirical observation shows that $\rho_{DM}$ in *E. coli* decreases from 0.31g/mL to 0.28g/mL as $\mu$ increases from 0.7h$^{-1}$ to 1.2h$^{-1}$ (S8A Fig and S1A Table) [10], corresponding to a reduction in occupancy from $\rho = 0.215$ to $\rho = 0.191$, an 11% decrease (S8B Fig and S1B Table). Thus, the predicted 15% change in $\rho_{opt}$ also appears to be consistent with the empirically observed reduction in occupancy in the transition to ultrafast growth.

Our simulations also show that even if the cell remains at the state of optimal occupancy, a large reduction in the nutrient level results in only a small decrease of the ribosome's saturation with its substrate (0.81 to 0.72, S5B Fig); at the same time, the ribosome's Michaelis parameter $K_M^*$ increases by 51% (S5C Fig). This finding is consistent with experimental observations that changes in translation rate per ribosome in *E. coli*, a direct consequence of ribosomal substrate saturation, are much smaller than the simultaneous changes in growth rate [38]. At increasing nutrient levels, a higher growth rate $\mu$ is facilitated by relocating a substantial amount of the ribosome's synthesis capacity from transporter proteins to metabolic enzymes and additional ribosomes (S4C Fig).

## Discussion

The linear pathway model shows that reactions with larger catalysts and substrates favor a lower occupancy than reactions with smaller molecules (Fig 3). This effect explains the observation of lower optimal occupancies for decreasing pathway length $N$ in the GBA model cell: the decrease in pathway length simulates the switch from minimal media, requiring on the order of 260 metabolic enzyme species to convert a small number of nutrients to the full range of cellular building blocks, to increasingly richer media, where progressively more biomass components can be taken up directly from the environment, requiring as few as 140 metabolic enzyme species. With decreasing numbers of active metabolic reactions $N$, the ribosomal sector expands at the expense of the metabolic sector, pushing $\rho_{opt}$ to lower values (S4A Fig). Given the minimalistic nature of the whole-cell model, which assumes that all metabolic reactions follow identical kinetics and approximates protein production through a single Michaelis-Menten type reaction, it is striking that the model not only predicts differences across physiological states, but predicts experimentally observed values [10] quantitatively with an error below 12%.

The whole-cell model predicts that the growth rate $\mu$ decreases only mildly when $\rho$ deviates from $\rho_{opt}$; for example, $\mu$ decreases by only 3% when $\rho$ increases to twice the optimal occupancy $\rho_{opt}$ (Fig 4; $N = 150$, $s_{ext} = 1.0$μM). This observation is consistent with an experiment that arrested the volume growth of a yeast cell while cytosolic dry mass continued to accumulate, increasing the concentration of a fluorescent protein—a proxy for dry mass density—to roughly twice the wildtype value [39]. These results suggest that the dry mass accumulation rate largely remains constant throughout the volume growth arrest, even when the cytosol density reaches twice the wildtype level.

In the whole-cell model, a 10% deviation of $\rho$ from $\rho_{opt}$ results in a 0.02% drop in $\mu$ (Fig 4); using growth rate as a proxy for fitness, this corresponds to a selection coefficient $s = 2$x$10^{-4}$. The effective population size of most bacterial species is on the order of $N_e = 10^8$ [40], and we thus have $s \gg 1/N_e = 10^{-8}$; accordingly, natural selection would be sufficiently strong to explain the difference in average $\rho_{DM}$ observed between the two physiological states. Experiments show substantial between-cell variation in each nutritional condition (S8 Fig) [10]: the

observed dry mass densities show coefficients of variation (standard deviation / mean) of around 5%. According to the whole-cell model, the corresponding difference in occupancy corresponds to a selection coefficient of $s = 10^{-4} \gg 1 / N_e$, indicating that this large cell-to-cell variation is not selectively neutral but persists despite negative selection arising from selection on rapid growth.

Thus, while our deterministic model predicts some variability, it does not predict variation of the observed magnitude. It thus appears that additional factors beyond selection for fast growth modify the density of individual cells either actively, through regulation, or passively, through the coupling of mass density to other factors. However, these observations do not invalidate our model, but rather indicate that other factors can be seen as perturbations of the state optimal for rapid growth, because our results are consistent with the average densities of cells across different growth conditions. One such influence may arise from the coupling of cell size to DNA replication. As growth rate increases, the number of replication rounds within a cell also increases (see Fig 9 of Ref. [41] for an illustration), and additional intracellular volume may simply be required to accommodate the parallel replication processes.

Questions about the optimal allocation of protein resources can be addressed by maximizing the growth rate (or flux) in computational models of cellular growth with fixed, crowding-unaware kinetic parameters. While such simulations produce meaningful predictions for the relative amounts of the proteins, they cannot limit absolute protein concentrations [42]. To solve this problem, existing models [13,14] implement hard, phenomenological constraints on the total concentration of protein or cellular dry mass, based on experimental observations that found these to be approximately constant across growth conditions [10,37,43,44]. Our results elucidate the biophysical origin of these observations, indicating that the cellular dry mass density represents a compromise between the saturation of metabolic enzymes with their substrates and the effects of reduced diffusion on the effective affinity of the ribosome for its much larger substrate, the ternary complex.

## Methods

### Crowding-adjusted Michaelis-Menten kinetics

Macromolecular crowding affects the flux of a metabolic reaction in multiple ways. It can (i) slow down diffusion; (ii) affect the free energy of substrate, catalyst, and the substrate-catalyst complex and thereby change their relative equilibrium ratios; and (iii) disturb the folding of a protein and affect the shape of the active site. In our modelling framework, we followed the derivation proposed in Minton [25] that systematically accounts for the effects of crowding on metabolic fluxes caused by effects (i) and (ii).

In this section, let us consider the metabolic reaction carried out by an enzyme $E$ that converts substrate $S$ into product $P$, in the presence of other volume-excluding co-solutes that, collectively, constitute the dry mass of the solution. The metabolic reaction is described by the chemical equation following Michaelis-Menten kinetics:

$$\text{E} + \text{S} \underset{k_{\text{off}}}{\overset{k_{\text{on}}}{\rightleftharpoons}} \text{ES} \overset{k_{\text{cat}}}{\longrightarrow} \text{E} + \text{P}$$

It is described by two parameters: the enzyme-substrate dissociation (or Michaelis) parameter $K_M \sim k_{\text{off}}/k_{\text{on}}$, and the catalytic rate constant (or turnover number) $k_{\text{cat}}$. Note that while $K_M$ is usually assumed to be invariable and is hence referred to as "Michaelis constant", we here examine crowding-dependent changes in $K_M$ und hence refer to it as the "Michaelis parameter". $K_M$ is sometimes defined as $K_M = (k_{\text{off}}+k_{\text{cat}})/k_{\text{on}}$. However, we here use the definition $K_M = k_{\text{off}}/k_{\text{on}}$, as this is the appropriate form when working with complex enzyme reactions [45].

For example, in a metabolic reaction with two substrates, A and B, the binding of the substrates to the enzyme involves kinetic parameters $k_{on}^A$, $k_{off}^A$, $k_{on}^B$, and $k_{off}^B$. The concentration of the enzyme complex, formed after both substrates have bound to the enzyme, can be described by $K_M^A = k_{off}^A / k_{on}^A$ and $K_M^B = k_{off}^B / k_{on}^B$ without directly involving the on- and off-rates. Thereafter, the flux of the reaction can be calculated by multiplying the complex concentration with $k_{cat}$. Inclusion of $k_{cat}$ into $K_M^A$ and $K_M^B$ complicates the derivation. For this reason, $k_{cat}$ is not included in $K_M$ in Minton [25], and we followed this convention in the current study.

We assume that all reactions follow effectively irreversible Michaelis Menten kinetics and thus free energy changes are irrelevant to the transition rate $k_{cat}$; moreover, we ignore crowding effects on the enzyme structure, and thus $k_{cat}$ is not affected by crowding. To derive the effect of crowding on the Michaelis parameter $K_M$, we divide a catalytic reaction into two independent, consecutive steps: (1) $S$ and $E$ diffuse until they meet, and then (2) $S$ and $E$ bind and unbind reversibly until the reaction proceeds forward to make $P$.

## Step (1): The substrate-catalyst encounter

The encounter rate between $S$ and $E$ depends on the cytosolic occupancy. Its rate law is similar to that of a diffusion-limited catalytic reaction, in which the $ES$ encounter rate is low and their encounter is the rate determining step; ys soon as the $ES$ complex is formed, the reaction quickly proceeds, converting $S$ into $P$ and releasing $E$. At given concentrations of $E$ and $S$, the rate of formation of $ES$ is thus mostly determined by the rate of encounter, which is proportional to the sum of $E$'s and $S$'s diffusion coefficients. A reduction of the diffusion rate shifts the equilibrium between the concentrations of the individual molecules and the complex $ES$; this can be accounted for in the following way [25]:

$$\frac{[ES]}{[E][S]} = \frac{1}{K_M^{diff}} \simeq \frac{1}{K_M^0} exp(-g(r_S)\rho); \tag{S1}$$

here, $K_M^{diff}$ is the hypothetical diffusion limited Michaelis parameter, while $K_M^0$ is the Michaelis parameter in the low-crowding limit; $\rho$ is the volume occupancy of the volume-excluding co-solutes (dry mass) of the solution (with range $0 < \rho < 1$), and g is a function that depends on the shape of E, S, and other volume-excluding co-solutes. Since S is typically much smaller than E, the diffusion coefficient of S in a crowded solution is in general much higher than that of E. Hence, we estimate this scaling term $exp(-g\rho)$ solely from the diffusion coefficient of S. Approximating S as a sphere of radius $r_S$, we can write it as $exp(-g(r_S)\rho)$.

The bacterial cytosol is crowded, slowing down the diffusion of all molecular species. The extent of this slow-down, however, is non-uniform and depends largely on the size of the affected molecule: the larger the molecule, the more it is slowed down. The slow-down of diffusion in the *E. coli* cytosol is summarized by the following empirical scaling law, which was inferred from molecular dynamics simulations [23]:

$$ln\left(\frac{D_0(r_h)}{D_{cyto}(r_h)}\right) = \left(\frac{\xi^2}{R^2} + \frac{\xi^2}{r_h^2}\right)^{\frac{-a}{2}} \tag{S2}$$

where $r_h = 1.3(r + 1.4\text{Å})$ is the hydrodynamic radius of a molecule with radius $r$ [46], *i.e.*, its effective radius including the attached water molecules; $D_0(r_h)$ is the diffusion coefficient in the low crowding limit, while $D_{cyto}(r_h)$ is the diffusion coefficient in the crowded cytosol condition; $\xi = 0.51$nm is the average distance between the surfaces of volume-excluding co-solutes in the cytosol of *E. coli*; $R = 42$nm is the radius of the largest common crowders in the cytosol; and $a = 0.53$ is an empirical scaling factor [23]. Note that in general, the parameter $\xi$ depends

on $\rho$; however, as we are only interested in the relationship between reaction fluxes, growth rate, and $\rho$ in a small range centered around the native *E. coli* cytosolic density, we can approximate $\xi$ by a constant in our analysis. As the cytosolic dry mass density is ~0.3g/mL [10] and the mass-to-volume-ratio of protein is 1.35g/mL [47], the cytosolic volume occupancy $\rho$ is approximately 0.22 = 0.3/1.35. If the reaction rate is proportional to the rate of encounter between *E* and *S*, and this rate of encounter is in turn approximately proportional to the diffusion coefficient of *S*, then Eq (S2) can be used to calculate the scaling factor $g(r_S)$:

$$g(r_s) = \frac{\left(\frac{\xi^2}{R^2} + \frac{\xi^2}{\left(1.3\left(r_S + 1.4\text{Å}\right)\right)^2}\right)^{\frac{-a}{2}}}{0.22}$$

and the rate of this reaction is denoted as $k_{\text{diff}}$:

$$k_{\text{diff}} = k_{\text{cat}}[ES] = \frac{k_{\text{cat}}[E][S]}{K_{\text{M}}^{\text{diff}}}$$

## Step (2): Repeated binding and unbinding of substrate and catalyst

The rate law of this step is related to a transition-state limited catalytic reaction, in which the rate of encounter of E and S is much larger than the rate of conversion of the *ES* complex into the product P. In this case, the complex *ES* exists in near equilibrium with *E* and *S*, and macromolecular crowding affects the reaction rate parameter mainly through shifting this equilibrium. To assess the magnitude of this effect, we consider the reversible reaction

$$\text{E} + \text{S} \underset{k_{\text{off}}}{\overset{k_{\text{on}}}{\rightleftharpoons}} \text{ES}$$

and denote the adjusted equilibrium Michaelis parameter in this hypothetical transition-limited case as $K_{\text{M}}^{\text{ts}}$ [25]:

$$K_{\text{M}}^{\text{ts}} = \frac{\gamma_E \gamma_S [E][S]}{\gamma_{ES}[ES]} = \Gamma \frac{[E][S]}{[ES]} = \Gamma K_{\text{M}}^0,$$

where $\gamma_i$ denotes the activity coefficient of molecular species $i$, $\Gamma = \frac{\gamma_E \gamma_S}{\gamma_{ES}}$, and $K_{\text{M}}^0 = \frac{k_{\text{off}}}{k_{\text{on}}}$ is the enzyme-substrate dissociation parameter in the low-crowding limit. The activity coefficients are defined as

$$\gamma_i = exp\left(\frac{\mu_i - \mu_i^{\text{ideal}}}{RT}\right)$$

with

$$\mu_i = \frac{\partial G}{\partial [i]};$$

here, G is the Gibbs free energy of the system; $[i]$ is the concentration of molecular species $i$; $\mu_i$ is the chemical potential of $i$, and $\mu_i^{\text{ideal}}$ is the chemical potential in an idealized situation, i.e., without intermolecular interactions and in the absence of other volume-excluding co-solutes. In other words, the equilibrium of the reaction will be identical to the dissociation parameter $K_{\text{M}}^0$ if the system is ideal. The $\Gamma$ term, therefore, accounts for the deviation of the Gibbs free energy from the idealized situation.

The $\gamma_i$ of each molecular species $i$ can be written as an expansion in terms of the concentrations of all molecular species [48]:

$$\ln \gamma_i = \sum_j B_{ij}[j] + \sum_{j,k} B_{ijk}[j][k] + \cdots\cdots.$$

The coefficients $B_{ij}$ ($B_{ijk}$,...) reflect the interaction between 2 (3,...) molecular species. For example, the coefficient $B_{ij}$ is given as [49]:

$$B_{ij} = 4\pi N_A \int_0^\infty \left(1 - exp\left(\frac{-U_{ij}(r)}{kT}\right)\right) r^2 dr$$

where $N_A$ is Avogadro's number, $r$ is the distance between the center of mass of molecular species $i$ and $j$, and $U_{ij}(r)$ is the potential of average force acting between the two molecular species. While the interaction potential among multiple molecular species is complex, it has been found that the colligative properties of solutions of globular proteins can be well accounted for over a wide range of concentrations by using a simple hard sphere potential, where two molecules cannot overlap but do not interact otherwise (see Minton [25] for a review):

$$U_{ij}(r) = \begin{cases} \infty & r \leq r_i + r_j \\ 0 & r > r_i + r_j \end{cases}$$

The scaled particle theory applies this rigid sphere potential to calculate the activity coefficient $\gamma_i$ of molecular species $i$ (Boublík, 1974):

$$\begin{aligned} ln\gamma_i = -ln(1 - \langle\langle V\rangle\rangle) \quad &+ \frac{r_i\langle\langle S\rangle\rangle + S_i\langle\langle r\rangle\rangle + V_i\langle\langle 1\rangle\rangle}{1 - \langle\langle V\rangle\rangle} \\ &\frac{+r_i^2\langle\langle S\rangle\rangle^2 + 2V_i\langle\langle r\rangle\rangle\langle\langle S\rangle\rangle}{2(1 - \langle\langle V\rangle\rangle)^2} + \frac{V_i\langle\langle r\rangle\rangle^2\langle\langle S\rangle\rangle^2}{3(1 - \langle\langle V\rangle\rangle)^3} \end{aligned} \tag{S3}$$

Here, $w_i$ is the concentration (number density) of molecular species $i$; $\langle\langle X\rangle\rangle = \sum_i w_i X_i$ is a weighted sum of property $X$, with $\langle\langle 1\rangle\rangle = \sum_i w_i$; $S_i = 4\pi r_i^2$ is the surface area and $V_i = \frac{4}{3}\pi r_i^3$ is the volume of molecular species $i$; in addition, $\langle\langle r\rangle\rangle = \sum_i w_i r_i$ and $\langle\langle r\rangle\rangle^2 = \sum_i w_i r_i^2$. As in the hypothetical transition-state limited case, the reaction rate parameter is proportional to the concentration of the enzyme-substrate complex, and a shift of its equilibrium constant by $\Gamma$ leads to a corresponding shift of the reaction rate, quantified by the rate parameter $k_{ts}$ [25]:

$$k_{ts} = k_{cat}[ES] = \frac{k_{cat}}{K_M^{ts}}[E][S] \tag{S4}$$

## Combining step (1) and (2) for the general formulation

We assume that a catalytic reaction is divided into two independent steps that occur in tandem: (1) the encounter of $S$ and $E$, whose rate law is proportional to that of a diffusion limited reaction, and thereafter (2) their reversible binding and unbinding until $S$ is converted into $P$, whose rate law is proportional to a transition-state limited reaction. The crowding-adjusted reaction rate $k$ is obtained by adding the inverse rate parameters (i.e., the reaction times) of the two steps [25,30,50]:

$$k^{-1} = k_{ts}^{-1} + k_{diff}^{-1}.$$

Given Eqs (S1) and (S4), this leads to [30]:

$$k = k_0 \frac{(1+\theta)\Gamma exp(-g\rho)}{\Gamma + \theta exp(-g\rho)}$$

Here, $\theta \equiv \frac{k_{ts}}{k_{diff}}\big|_{\rho=0}$ quantifies the relative contributions of step (1) and (2) to the overall reaction, or in other words the ratio between times spent on step (1) and on (2) at low cytosolic occupancy; the reaction tends towards diffusion limited if $\theta \rightarrow 0$, or transition-state limited if $\theta \rightarrow \infty$. $k_0$ is the overall rate parameter in the absence of crowding. Remember that $[E_{free}] = [E_{total}] - [ES] = [E_{total}]/(1 + [S]/K_M)$ is the concentration of free enzymes only, and thus the reaction rate is $k = k_{cat} [ES] = k_{cat} [E_{free}] [S]/K_M$. Let us also denote the total and free substrate concentration as $[S_{total}]$ and $[S] \equiv [S_{free}]$, which are connected by the relationship $[S_{free}] = [S_{total}] - [ES]$. But because $[S_{total}] \gg [ES]$, we have $[S_{total}] \cong [S_{free}]$ and can ignore the difference between $[S_{total}]$ and $[S_{free}]$ in the equations. From now on, we describe the reaction rate as the flux of the reaction, and denote it as $v$. As we assume that the reaction is irreversible and hence crowding does not influence $k_{cat}$, to arrive at crowding-adjusted Michaelis-Menten kinetics, we have to scale the Michaelis parameter, which arises from considerations on the equilibrium between the enzyme-substrate complex and its constituents. Thus, $v/v^0 = K_M^0/K_M^*$, with $K_M^0$ the Michaelis parameter in the low crowding limit; then the crowding-adjusted Michaelis parameter is

$$K_M^* = K_M^0 \frac{\Gamma + \theta exp(-g\rho)}{(1+\theta)\Gamma exp(-g\rho)} \tag{S5}$$

and so the flux can be written as

$$v = k_{cat} \frac{[E_{free}][S]}{K_M^*} \simeq k_{cat} \frac{[E_{total}][S]}{K_M^* + [S]}. \tag{S6}$$

The $\theta$ of ERK MAP kinase phosphorylation reaction was estimated to be 2.3 [30]; we assume that this value is representative for cellular enzymes and use it for modeled reactions. To understand how the choice of $\theta$ affects the model behavior, we also ran a model where metabolic reactions have $\theta = 4.6$.

The concentration of the enzyme-substrate complex, $[ES]$, depends on $K_M^*$ through $[ES] = [E_{total}][S]/([S] + K_M^*)$; at the same time, $K_M^*$ depends on the concentration of different molecular species, including $[ES]$, through Eq (S3). To find a self-consistent solution for these two quantities, we iterate these two equations until convergence. Initially, we are given the total concentration of the enzyme and substrate, $[E_{total}]$ and $[S_{total}]$, which are invariant throughout the iterations. The level of crowding is specified by three molecular species— $[E_{free}]$, $[S_{free}]$, and $[ES]$; let us also denote $[E_{free}^n]$, $[S_{free}^n]$, and $[ES^n]$ to represent their corresponding values in the $n^{th}$ iteration step. At the initial interaction step ($n = 0$), we set $[E_{free}^0] = [E_{total}]$, $[S_{free}^0] = [S_{total}]$, and $[ES^0] = 0$. At iteration $n$, we first calculate $K_M^{*n}$ based on $[E_{free}^n]$, $[S_{free}^n]$, and $[ES^n]$, and thereby calculate $[E_{free}^{n+1}]$, $[S_{free}^{n+1}]$, and $[ES^{n+1}]$ before proceeding to the next iteration $n+1$. We stop the iteration when all $K_M^*$ values changed by less than 0.001% compared to the previous iteration.

## Single-pathway model

We considered a simple model of a linear pathway to investigate how the size of the substrate and catalyst of a metabolic reaction affect the optimal cytosolic occupancy; here, optimality is defined as a maximal pathway flux per unit dry mass, calculated from crowding-adjusted

kinetics. The pathway is divided into $N$ steps, where $E_n$ is the catalyst of step $n$, which converts its substrate $s_n$ into $s_{n+1}$, the substrate of the next step (Fig 2A). We assume that all internal metabolite concentrations are in steady state (i.e., producing and consuming fluxes cancel exactly), and we ignore the dilution of intermediates. Thus, all reaction fluxes have the same value, $v$. We assume that $s_1$ is replenished by a flux $v_{s_1} = v$, which is not modeled explicitly.

We assume that the $N$ reactions are described by crowding-adjusted Michaelis Menten kinetics with identical $k_{\mathrm{cat}}$ and $K_M^0$. We further assume that the $N$ catalyst species and the $N$ substrate species are spherical, with radius $r_E$ (volume $V_E$) for the catalysts and radius $r_s$ (volume $V_s$) for the substrate species. These assumptions simplify the solution space of the model, as in the optimal steady state, all catalysts have equal total concentrations ($[E_1] = [E_2] = \ldots = [E_N]$, and so do all the substrates ($[s_1] = [s_2] = \ldots = [s_N]$). We define the total substrate concentration $[s] \equiv \sum_n [s_n] = N[s_1]$ and the total catalyst concentration $[E] \equiv \sum_n [En] = N[E_1]$. The two variables $[s]$ and $[E]$ span the solution space of this model. Substrates and catalysts, as well as the substrate-catalyst complexes, are crowders in their own right; they slow down diffusion and perturb Gibbs free energies. The cytosolic volume occupancy of dry mass in the solution, $\rho$ ($0 \leq \rho \leq 1$), is

$$\rho = N_A \frac{4\pi}{3} \left( [s]r_s^3 + [E]r_E^3 \right) \tag{S7}$$

with the Avogadro number $N_A$. The flux through the pathway per unit volume is

$$v = k_{\mathrm{cat}} \frac{\left(\frac{[s]}{N}\right)\left(\frac{[E]}{N}\right)}{K_M^* + [s]/N} = \frac{k_{\mathrm{cat}}}{N} \frac{[s][E]}{NK_M^* + [s]}$$

Accordingly, the flux per unit dry mass is

$$\mu = \frac{k_{cat}}{N} \frac{[s][E]}{NK_M^* + [s]} \frac{1}{\rho}. \tag{S8}$$

As we ignore crowding effects on the turnover number, $k_{\mathrm{cat}}$ acts only as a scaling factor, and we thus set $k_{\mathrm{cat}} = 1$ for simplicity. We set the Michaelis parameter at the low crowding limit to $K_M^0 = 130\mu$M, which is the median $K_M$ of metabolic enzymes [33], and is also very close to the Michaelis constant of the ribosome estimated from the diffusion limit without molecular crowding [34].

Because we assume identical kinetics of all reactions and ignore the dilution of intermediates through growth, the whole pathway is equivalent to a single reaction with re-scaled kinetics. We still describe it as an $N$-steps pathway, as this more faithfully reflects the situation in the real cell, allowing us to use realistic parameter values. Moreover, while mathematically, $N$ represents a scaling factor, it has an intuitive biological interpretation.

The same equations can be used to describe a system of $N$ parallel enzymatic reactions with identical fluxes, with only an additional multiplication by $N$ in Eq (S8), $\mu_{parallel} = \mu_N$ (Fig 2B). Here, catalyst $E_n$ of reaction $n$ converts substrate $s_n$ into the end product, and the consumption of $s_n$ is compensated by a flux $v_{s_n}$ that supplies $s_n$ at an equal rate.

We consider two systems of substrate and catalyst sizes: metabolic and ribosomal. In the metabolic system, we use $r_s = 0.34$nm for metabolites (the approximate radius of the amino acid alanine [51]) and $r_E = 2.4$nm for globular proteins (an "average" protein in *E. coli* has a mass 40kDa [3], while a globular protein with mass 50kDa has a radius $r = 2.4$nm [52]; in an alternative estimation, the radius of a typical globular protein is approximately $r = 2.5$nm [53]). In the ribosomal system, we use $r_s = 2.4$nm for the ternary complexes (the gyroscopic radius of tRNA is estimated to range from 2.33nm to 2.46nm based on Eq. (7) of Hyeon et al.

[54]). We use a radius of $r_E$ = 13nm for the ribosome, as the diameter of a ribosome is reported to be 20nm - 30nm [53, 55, 56]. We assume that the catalyst-substrate complex is also spherical, with a volume equal to the sum of the substrate's volume and the catalyst's volume. For both metabolic and ribosomal systems, we used crowding-adjusted Michaelis-Menten kinetics with $\theta$ = 2.3; to explore the effect of assuming the same $\theta$ value for both systems, we alternatively examined a model where we set $\theta$ to 4.6 for the metabolic system, as the metabolic system may have a higher diffusion efficiency than the ribosomal system.

For a fixed value of the total occupancy $\rho$, we calculated the specific flux $\mu$ using MATLAB, while (i) varying the occupancy $\rho$ in steps of 0.01 from 0.01 to 0.8, with additional, finer-grained steps of 0.001 from 0.100 to 0.360, and (ii) varying the ratio of the volume occupied by the substrates $s$, from 0.1% to 97.7%, with an increase by a factor of 1.0023 at each step.

### Single-pathway model: The Vazquez approach

In Vazquez [26], reactions are classified into two contrasting types: (1) those in the saturation limit, with $[S] \gg K_M$, and (2) those in the diffusion limit, with $[S] \ll K_M$. The rate of reactions in the diffusion limit is modified by an exponential term, $\exp(-5.8\rho)$, where $0 \leq \rho \leq 1$ is the cytosolic occupancy. In addition, crowding increases the contact with enzymes and substrates, and so the reactions are sped up by the term $1/(1 - \rho)$. Overall, the reaction rate of a reaction in saturation is

$$v_{\text{sat}} = k_{\text{cat}} \frac{1}{1 - \rho} \frac{[s][E]}{[s] + K_M},$$

while the reaction rate in the diffusion limit has an extra exponential term,

$$v_{\text{diff}} = k_{\text{cat}} \exp(-5.8\rho) \frac{1}{1 - \rho} \frac{[s][E]}{[s] + K_M}.$$

We simulated the metabolic and ribosomal systems based on these two equations, setting $k_{\text{cat}}$ = 1 and assuming the metabolic system to be in saturation and the ribosomal system to be in diffusion limit (S1 Fig).

### Model cell with a metabolic and a ribosomal pathway

To more faithfully represent a complete cell and to study the tradeoff between metabolic and ribosomal reactions, we also consider a balanced growth model with a metabolic sector and a ribosomal sector (Fig 2C). As seen from the results for the pathway models, the two types of reactions have very different optimal conditions: the metabolic sector involves smaller catalysts and substrates than the ribosomal sector, and hence has maximal specific fluxes at a higher cytosolic occupancy.

In this model (Fig 2C), the transporter $T$ imports the external substrate $s_{\text{ext}}$ into the cytosol, where it is now labeled $s_1$. The transport reaction is described by ordinary, crowding-unaware irreversible Michaelis Menten kinetics, with flux

$$v_T = k_{\text{cat}}^T [s_{\text{ext}}][T] / \left([s_{\text{ext}}] + K_M^T\right).$$

We set $k_{\text{cat}}^T = 13.7\text{s}^{-1}$ (the median $k_{\text{cat}}$ of enzyme reactions [33] (Bar-Even et al., 2011)), and $k_M^T = 1\mu\text{M}$ (close to the growth limiting glucose concentration of E. coli [57]).

The metabolic sector comprises an $N$-steps linear pathway of metabolic reactions, identical to the one studied in the simple pathway model: the enzyme of metabolic reaction $n$ ($1 \leq n \leq N$), denoted as $M_n$, converts substrate $s_n$ into $s_{n+1}$; in reaction $n = N$, $M_N$ converts one $s_N$ into

one precursor $p$ [12]. These $N$ reactions follow crowding-adjusted irreversible Michaelis Menten kinetics with identical rate parameters $k_M^{M0} = 130\mu M$ and $k_{cat}^M = 13.7 s^{-1}$ (the median $K_M$ and $k_{cat}$ of enzyme reactions [33]). As before, all $N$ substrates have radius $r_s = 0.34nm$, while all $N$ enzymes have radius $r_M = 2.4nm$; as before, we assume that the enyzme-substrate complex is spherical and occupies as much volume as one substrate plus one enzyme molecule. To facilitate the numerical solution of the model, we assume that all metabolite concentrations $[s_n]$ and also all enzyme concentrations $[M_n]$ are identical. This corresponds to the optimal balanced growth solution when the differential dilution of intermediate metabolites is ignored [33]; thus, we treat the dilution of intermediate metabolites only approximately here. In the balanced growth condition, the production rate of each substrate is equal to its consumption rate plus its (approximate) rate of dilution through growth, so that its concentration remains stable. We define the total substrate concentration $[s] \equiv \sum_n [s_n]$ and total enzyme concentration $[M] \equiv \sum_n [M_n]$. The flux of each metabolic reaction is $\frac{k_{cat}^M}{N} \frac{[s][M]}{N K_M^{M*} + [s]}$.

The ribosomal sector comprises the ribosome ($R$) and the protein precursor ($p$): $R$ converts $p$ into the $N+2$ types of protein in the model, $N$ metabolic enzymes ($M_n$), the ribosome ($R$), and the transporter ($T$). As before, the radius of $p$ is $r_p = 2.4nm$ and the radius of $R$ is $r_R = 13nm$; their complex is assumed to be spherical and to occupy as much volume as one precursor plus one ribosome molecule. The ribosomal conversion rate is described by crowding-adjusted irreversible Michaelis Menten kinetics with parameters $k_M^{R0} = 120\mu M$ [34] and $k_{cat}^R = 22.0 s^{-1}$ [12,34], and the consumption rate of $p$ by $R$ to make proteins is $k_{cat}^R \frac{[p][R]}{K_M^{R*} + [p]}$. The ribosome converts $l_T = 300$ precursors into one transporter, $l_M = 300$ precursors into one metabolic enzyme, and $l_R = 7459$ precursors into one ribosome [12].

Note that the reactions that produce and consume the precursor $p$ do not conserve volume. The reason is that to model a realistic cell, we envision the precursor as a charged tRNA, only the amino acid part of which is (i) produced by the metabolic pathway and (ii) integrated into the growing protein. The metabolic pathway only provides the amino acid, while the pool of free tRNAs is not explicitly modeled. For this reason, the size of $p$ is substantially larger than the size of the metabolite $s_N$ consumed in its production. Conversely, the ribosome consumes 300 precursors to produce a single transporter or enzyme, which are both substantially smaller than the combined volume of the 300 precursors. Here, we envision that the tRNA part of $p$ is set free again and can be re-charged through $M_N$. This treatment assumes that the concentration of free tRNA is so low that we can ignore its dilution through growth and its contribution to the cytosolic crowding.

The solution space of this model cell spans five dimensions: $[s]$, $[p]$, $[T]$, $[M]$, and $[R]$. The corresponding molecules, along with their complexes, are also the crowders that slow down diffusion and disturb Gibbs free energies. The cytosolic occupancy $\rho$, $0 \le \rho \le 1$, is

$$\rho = N_A \frac{4\pi}{3} \left( [s] r_s^3 + [M] r_M^3 + [p] r_p^3 + [R] r_R^3 \right) \tag{S9}$$

with the Avogadro number $N_A$. The growth rate $\mu$ of this model cell can be expressed as the flux through the ribosome reaction divided by the total protein concentration,

$$\mu = k_{cat}^R \frac{[p][R]}{K_M^{R*} + [p]} \frac{1}{N_A (l_M [M] + l_R [R] + l_T [T])} \tag{S10}$$

In the balanced growth state, the production of $s$ and $p$ is offset by their consumption and dilution by growth,

$$k_{\mathrm{cat}}^{\mathrm{T}} \frac{[s_{ext}][T]}{K_{\mathrm{M}}^{\mathrm{T}} + [s_{ext}]} - \frac{k_{\mathrm{cat}}^{\mathrm{M}}}{N} \frac{[s][M]}{NK_{\mathrm{M}}^{\mathrm{M*}} + [s]} - \mu[s] = 0$$

$$\frac{k_{\mathrm{cat}}^{\mathrm{M}}}{N} \frac{[s][M]}{NK_{\mathrm{M}}^{\mathrm{M*}} + [s]} - k_{\mathrm{cat}}^{\mathrm{R}} \frac{[p][R]}{K_{\mathrm{M}}^{\mathrm{R*}} + [p]} - \mu[p] = 0 \tag{S11}$$

For a given number of enzymes $N$ and occupancy $\rho$, we solved this model numerically. As a preliminary step, we used the BARON algorithm [58] implemented in Pyomo [59,60] and assumed normal crowding-unaware irreversible MIchaelis Menten kinetics, maximizing the growth rate $\mu$ over the space of concentrations ($[s],[p],[T],[M],[R]$), subject to the constraints of Eqs (S9) and (S11). In the main model, we assumed $\theta = 2.3$ for both metabolic and ribosomal reactions. As an alternative, we also examined a model with metabolic reactions biased more toward the transition state limit, setting $\theta = 2.3$ for ribosomal reactions and $\theta = 4.6$ for metabolic reactions.

Using the BARON solution as a starting point, we applied the SLSQP algorithm within the function "*minimize*" in SciPy [61], now using crowding-adjusted Michaelis-Menten kinetics. It is not clear if the optimization problem has a unique solution; to increase the probability that the solution is a global maximum, we repeatedly ran the SLSQP algorithm at least 20 times for each ($N,\rho$) combination, and picked the solution with the highest $\mu$. For each simulation, at least half of the independent runs supported the same, maximal optimum.

## Estimating the number of enzymes in the metabolic pathway

To obtain a realistic estimate of the number of simultaneously active metabolic reactions in a bacterial cell, we performed flux balance analysis simulations accounting for molecular crowding in terms of a hard limit on the total cellular protein concentration. Simulations were run using "sybil", an R library for efficient constraint-based analyses [36,62,63]. We used sybilccFBA [64,65], a re-implementation of the MOMENT algorithm with an improved treatment of multifunctional enzymes, which maximizes the biomass production rate while constraining the sum of cytosolic metabolic enzyme concentrations at the experimentally observed level.

We used a sybilccFBA implementation of the iAF1260 stoichiometric model [66], parameterized with turnover numbers for *E. coli* [64,65]. We considered four different nutritional environments. In each case, we counted the number of active metabolic reactions with both substrates and products located in the cytosol, and with a flux $>10^{-6}$ mmol/(gram Dry Weight)/h to filter out numerical noise; we enumerated the enzymes supporting these reactions, filtering out those with proteome fraction lower than a cutoff ($10^{-9}$; for reference, the most abundant enzyme has a density $\sim 10^{-3}$).

The estimated numbers of active metabolic enzymes and reactions in the different conditions are as follows: (i) 259 active enzymes and 349 active reactions in a minimal medium with glucose as the sole carbon source, corresponding to slow to intermediate growth with a cytosol dominated by the metabolic sector [64,67] (S2 Table); (ii) 206 active enzymes and 288 active reactions in the same minimal glucose medium supplemented with 20 amino acids, simulating intermediate growth [64,67] (S3 Table); (iii) 174 active enzymes and 250 active reactions in a rich medium, corresponding to fast growth and a cytosol dominated by the ribosome and its substrates [64,68] (S4 Table); and (iv) 140 active enzymes and 234 active reactions in an

extremely rich medium, where all exchange reactions in the iAF1260 *E. coli* model are allowed to be active.

## Conversion of experimental dry mass density to occupancy

An empirical growth law connects the RNA/protein mass ratio ($r$) of an *E. coli* cell with its growth rate $\mu$: $r = 0.087 + \mu/(4.5h^{-1})$ (Eq. (1) of Scott et al. [1]); a further experiment reports the subtle deviation of this relationship from linearity (See Fig 1D of Dai et al. [38]). At $\mu = 0$ and $0.7h^{-1}$, $r$ is measured to be 0.086 and 0.225 (S3 Table of Dai et al. [38]). The average specific density of protein is 1.35 g/mL [47], while that of the *E. coli* 70S ribosome is (1.637 g/mL), where RNA constitutes 61.87% and the remaining mass fraction is assumed to be proteins [69]. From these relationships, we can obtain the RNA density, resulting in a value of 1.81 g/mL.

If we ignore all molecular species in the dry mass except protein and RNA (which constitute ~75% of the dry mass in an *E. coli* cell [70–72]), we can estimate the overall specific density of dry mass (i.e., mass per volume of dry mass [g/mL], denoted as $D$) from $r$ using the following formula

$$D = 1.35 \frac{1}{1+r} + 1.81 \frac{r}{1+r} \tag{S12a}$$

And then $\rho$ can be estimated from $D$ and cytosolic dry mass density $\rho_{\text{DM}}$

$$\rho = \frac{\rho_{\text{DM}}}{D} \tag{S12b}$$

Applying Eq (S12a), we find that the overall specific density of dry mass $D$ increases from 1.39g/mL to 1.43g/mL when $\mu$ increases from ~0h$^{-1}$ to 0.7h$^{-1}$ *(r* changes from 0.86 to 0.243, reported from Dai et al. [38]). Oldewurtel et al. [10] found that up to these growth rates, the observed cellular dry mass density in *E. coli* is approximately constant at $\rho_{\text{DM}} = 0.31$g/mL. Plugging these numbers into Eq (S12b), the empirical $\rho$ decreases from 0.223 to 0.215, a 4% reduction. This shows that the cytosolic volume density $\rho$ changes even when the cytosolic mass density does not change.

The same set of experiments also showed that $\rho_{\text{DM}}$ in *E. coli* decreases from 0.31g/mL to 0.28g/mL as $\mu$ increases further, from 0.7h$^{-1}$ to 1.2h$^{-1}$ [10]. Within this range of $\mu$, $r$ increases from 0.225 to 0.33 ($r = 0.33$ when $\mu = 1.17$h$^{-1}$; S3 Table of Dai et al. [38]), and $D$ changes from 1.43g/mL to 1.46g/mL (based on $r = 0.33$) [47,70,73]. Accordingly, the empirical occupancy $\rho$ decreases from 0.215 to 0.191, a decrease of 11%. S8 Fig shows the boxplot of the distribution of $\rho_{\text{DM}}$ and $\rho$ in different nutritional environments. It shows a trend of roughly constant $\rho_{\text{DM}}$ but decreasing $\rho$ with increasing $\mu$ at low growth conditions. S1A and S1B Table show the tests that compare the distributions of $\rho_{\text{DM}}$ and $\rho$, respectively, between different nutritional conditions.

## Supporting information

**S1 Fig. (A) reaction fluxes and (B) growth rate (flux per unit dry mass) of the linear model based on the treatment of crowding effects by Vazquez 2010.** Metabolic reactions (blue curves), which are assumed to be saturated by the substrates, do not display any optimal cytosolic density at any intermediate values. In contrast, ribosomal reactions (red curves), which are assumed to be in the diffusion limit, show optimal densities. Solid lines are for smaller systems with $N = 20$ reactions, dashed lines are for larger systems of $N = 100$ reactions. (DOCX)

**S2 Fig. The optimal cytosolic occupancy increases with metabolic pathway length $N$ and decreasing external nutrient concentration $s_{\text{ext}}$.** This applies to a modified whole cell model with more diffusion efficient metabolic reactions ($\theta = 4.6$ for metabolic reactions and $\theta = 2.3$ for ribosomal reactions; thick solid curves) compared with the main model ($\theta = 2.3$ for both types of reactions; thin broken curves; replicates of Fig 5).
(DOCX)

**S3 Fig. The dependence on the number of active metabolic reactions $N$ for the volume fraction (A) and concentration (B) of substrate $s$, protein precursor $p$, transporter $T$, metabolic enzyme $M$, and ribosome $R$, and the proportion of ribosomal activities dedicated to the three types of proteins (C).** Each column of plots shows data for a different nutrient concentration in the environment, $s_{\text{ext}}$ (in μM).
(DOCX)

**S4 Fig. The volume fraction (A), substrate saturation (B), and $K_{\text{M}}^{*}$ (C) of the metabolic sector (blue) and ribosomal sector (red), plotted against $N$.** Each column of plots shows data for a different nutrient concentration in the environment, $s_{\text{ext}}$ (in μM).
(DOCX)

**S5 Fig. The volume fraction (A), saturation (B) and crowding-adjusted Michaelis parameter $K_{\text{M}}^{*}$ (C) of the metabolic sector (blue) and ribosomal sector (red), plotted against different nutrient concentrations in the environment, with fixed pathway length $N = 250$, at optimal occupancy (solid curves), and with fixed occupancy $\rho = 0.22$ (broken curves).**
(DOCX)

**S6 Fig. The volume fraction (A) and concentration (B) of substrates $s$, precursor $p$, transporter $T$, metabolic enzymes $M$, and ribosome $R$, and also the proportion of ribosomal activities dedicated to the three types of proteins (C), plotted against different nutrient concentrations in the environment, with $N$ fixed at 250 (solid curves), and with $N$ fixed at 250 and $\rho$ fixed at 0.22 (dashed curves).**
(DOCX)

**S7 Fig. Opposing effects of molecular crowding on the Michaelis parameter $K_{\text{M}}^{*}/K_{\text{M}}^{0}$ through perturbations of diffusion and of Gibbs free energies.** Simulations considered the individual and combined effects of diffusion limitation and transition state limitation for a parallel system of ribosomal reactions (red) and for a linear system of metabolic reactions (blue: $\theta = 2.3$; green: combined effect with a stronger bias towards transition state limitation, $\theta = 4.6$). **(A)** $N = 20$ and **(B)** $N = 100$.
(DOCX)

**S8 Fig. Boxplot displaying the distribution of (A) cytosolic mass density ($\rho_{\text{DM}}$) and (B) occupancy ($\rho$) across different nutritional environments.** Symbols on the x-axis labels are; MM: minimal medium; man: mannose; gly: glycerol; glu: glucose; CAA: casamino acids; RDM: rich dry medium. To limit the y-axis range, both panels do not show the outlier points. Each bar in **(A)** corresponds to the distribution of $\rho_{\text{DM}}$ measurements of wildtype *E. coli* cells (MG1655) cultured in the same nutritional condition, reported in Oldewurtel et al. 2021. With the exception of MM+gly, all conditions of minimal media have indistinguishable mean cytosolic mass density (See S1A Table for the results of the tests). Each bar in **(B)** corresponds to the $\rho$ estimated from the $\rho_{\text{DM}}$ measurements of the same condition using Eq. (12a) and Eq. (12b). $r$, the RNA/protein mass ratio, is necessary in the calculation of $\rho$; we estimated $r$ from the growth rate $\mu$ using the MATLAB interpolation function 'interp1' and the $\mu$-$r$ measurements of wildtype *E. coli* in S3 Table of Dai et al. 2016. In contrast to **(A)**, all distributions

here have different mean values (See S1B Table for the results of the statistical tests). Excluding MM+gly, this graph shows a trend of decreasing $\rho$ with increasing $\mu$.
(DOCX)

**S1 Table. Pairwise comparisons, based on two-sided Wilcoxon rank-sum tests, of cytosolic mass density $\rho_{DM}$ (A) and cytosolic occupancy $\rho$ (B) of wildtype *E. coli* (MG1655) cells cultured in different nutritional conditions.** $\rho$ is calculated from $\rho_{DM}$ and $r$, the RNA/protein mass ratio, using Eq. (12a) and Eq. (12b). $r$ is estimated from the $\mu$ using the MATLAB interpolation function 'interp1' and the $\mu$-$r$ measurements of wildtype *E. coli* reported in S3 Table of Dai et al. 2016. Each cell in the two tables corresponds to the p-value of the Wilcoxon rank-sum test, colored with red (significantly different / dissimilar) and blue (not significantly different / similar) with cutoff significance level 0.05. Symbols of the condition labels are; MM: minimal medium; man: mannose; gly: glycerol; glu: glucose; CAA: casamino acids; RDM: rich dry medium. S1A Table. P-values of spearman correlation to compare the cytosolic dry mass density $\rho_{DM}$ between different nutritional conditions. S1B Table. P-values of spearman correlation to compare the cytosolic occupancy $\rho$ between different nutritional conditions.
(DOCX)

**S2 Table. List of metabolites for the glucose only nutritional environment; it is a reproduction of S3 Table in page 97 of Alzoubi 2019, excluding the 20 amino acids.**
(DOCX)

**S3 Table. The nutrients in the environment for the glucose + 20 amino acids medium includes those listed in S1 Table, plus the following 20 amino acids.**
(DOCX)

**S4 Table. The list of nutrients in the environment of the rich medium; it is a reproduction of S4 Table in page 98 of Alzoubi 2019, excluding biotin, pyridoxine, selenate and selenite as their exchange reactions are missing in the iAF1260 model.**
(DOCX)

## Acknowledgments

We thank Deya Alzoubi and David Heckmann for providing the environments used in the ccFBA simulations. We thank Hugo Dourado and Deniz Sezer for helpful discussions.

## Author Contributions

**Conceptualization:** Tin Yau Pang, Martin J. Lercher.

**Formal analysis:** Tin Yau Pang.

**Investigation:** Tin Yau Pang, Martin J. Lercher.

**Methodology:** Tin Yau Pang.

**Software:** Tin Yau Pang.

**Writing – original draft:** Tin Yau Pang, Martin J. Lercher.

**Writing – review & editing:** Tin Yau Pang, Martin J. Lercher.

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
