## [Decision Letter · Decision Letter 0]

20 Dec 2022

Dear Prof Lercher,

Thank you very much for submitting your manuscript "Optimal density of bacterial cells" for consideration at PLOS Computational Biology.

As with all papers reviewed by the journal, your manuscript was reviewed by members of the editorial board and by several independent reviewers. In light of the reviews (below this email), we would like to invite the resubmission of a significantly-revised version that takes into account the reviewers' comments.

As you can see, the two reviewers make several critical comments and suggestions which you should address. In particular, reviewer 1 raises the question of experimental support and specifically whether the observed density difference between fast and slow growth is meaningful given the observed variability, which is an important point. Without experimental support, the appeal of the theoretical approach is considerably lower. A second point of reviewer 1 concerns the role of size control here with size effectively being volume. From my reading, it seems this is not essential here, because the density enters the model as a conversion factor between proteome fractions (relevant for resource allocation) and concentration (which enter rates), but I may be misunderstanding. In any case, clarifying this would be helpful

We cannot make any decision about publication until we have seen the revised manuscript and your response to the reviewers' comments. Your revised manuscript is also likely to be sent to reviewers for further evaluation.

Sincerely,

Stefan Klumpp

Academic Editor

PLOS Computational Biology

James O'Dwyer

Section Editor

PLOS Computational Biology

As you can see, the two reviewers make several critical comments and suggestions which you should address. In particular, reviewer 1 raises the question of experimental support and specifically whether the observed density difference between fast and slow growth is meaningful given the observed variability, which is an important point. Without experimental support, the appeal of the theoretical approach is considerably lower. A second point of reviewer 1 concerns the role of size control here with size effectively being volume. From my reading, it seems this is not essential here, because the density enters the model as a conversion factor between proteome fractions (relevant for resource allocation) and concentration (which enter rates), but I may be misunderstanding. In any case, clarifying this would be helpful

Reviewer's Responses to Questions

**Comments to the Authors:**

Reviewer #1: Molecular crowding in a cell affects various aspects of biochemical reactions. The paper proposes that the concentrations of molecules in bacterial cytosol are optimized to maximize the effect of molecular crowding on the growth rate under the constraint of resource allocation. The authors propose and analyze several versions of phenomenological models to simulate the effect based on rate equations that take into account the molecular crowding effect in the reaction rate. The models predict that a lower concentration is optimal for a higher growth rate where the higher fraction of the total proteins is the ribosomal proteins. The prediction is a few to 15 % reduction of change of the concentration within a physiologically reasonable change of the growth rate. The authors claim that the prediction is consistent with the experiential observation in reference [6] (Oldewurtel et al.).

While I don’t know the literature clearly against the proposed idea, I don’t find the mentioned experimental evidence convincing either (see below). The work is speculative, and even though I like speculative ideas in general, I find the paper lacks a critical evaluation of their proposal in comparison to other possibilities.

Major criticism:

1)The only comparison with experimental data made by the authors about the growth rate dependence of the bacterial cytosol density is with the data from Oldewurtel et al.. The paper measured the dry mass of bacterial cells at the single-cell level. As the authors also wrote in the discussion, the data have about a 5% of deviation. The authors state the data supports a 6% of reduction of the concentration between the very low growth rate (close to zero) to 0.7/hour, but it is not a meaningful difference with the given spread of data. In addition, the authors use the linear extrapolation of the ribosome fraction from Scott et al. [1] in this calculation. However, the data in the paper by Dai et al. that the authors also cite in [34] show a clear deviation from the linear fit in this range of growth rate. If the data in Dai et al. is used, the predicted difference will be even smaller even if one follows the authors' calculation method, making the prediction less relevant compared to the measurement precision. For a higher growth rate, the decrease is claimed to be 13% from the data and the model prediction is 15%. The authors only state that they are consistent and that I agree that they are not inconsistent, but I don’t see them as an agreement to support the theory.

2)There is a large body of research on how cell size is regulated in balanced growth in E. coli. Since the cell size (volume) and the content together give the density, I found it rather misleading that the current manuscript does not talk about the cell size volume at all. For example, the paper [6] states in the significance statement that ” First, cells expand surface area, rather than volume, in proportion with biomass growth. Second, cell width is controlled independently, with an important influence of turgor pressure.” These regulations then result in a certain dry mass density for a given growth rate. How is this related to the current paper’s point of view? Another body of literature that the current manuscript is not discussing at all is the coupling between cell size and DNA replication, reviewed e.g. in Jun et al. Rep. Prog. Phys. 81 (2018) 056601. In this view, the cell size is determined by the initiation mass per origin of replication. This does not directly tell the concentration of various macromolecules in the cell, but clearly indicates that the packing of DNA is a very important determinant to determine the volume of the cell. To me, it sounds unlikely that DNA replication plays no role in determining the density of molecules in the cell. For example, I could imagine a scenario where the cell volume needs to increase quite a lot for faster growth to accommodate the space for DNA replication, but making all the proteins at the same proportionality requires a quite large increase in the production rate of the proteins, that cell cannot catch up and density decrease with the growth rate - in other words, the decrease of the density is a side effect. The current manuscript does not discuss any argument against such possibilities. Also, while there may be limited literature that determined dry mass at the single cell level, there are many pieces of literature that have average numbers for macromolecules for various growth conditions (e.g. Bremer and Dennis), and that is all the authors need to compare their theory with, which is only about the mean values. Listing and discussing various data from the literature on growth rate dependence of the cell volume and macromolecule mass and giving a critical evaluation of their own proposal is a necessity when proposing a speculative idea.

A technical comment

Line 635 gives the translation rate in the model, and the maximum translation rate k_cat^R is set to 22.0/s. But this number is the measured ribosomal elongation rate, which is already affected by molecular crowding effects. Shouldn’t it be set so that the overall rate becomes consistent with this number after one puts in the crowding and the substrate concentration in the entire expression?

Reviewer #2: Pang and Lercher present an interesting study of how molecular crowding affects cellular growth rate. They base their analysis on a derivation of an effective Michaelis parameter that, compared to the original Michaelis constant K_m^0, accounts for the effects of crowding through a) changes in Gibbs free energy, and b) slow-down of diffusion. Different types of reactions (metabolic vs ribosomal) are parameterised through a relative weight \\theta, which is larger for reactions with smaller substrates that may diffuse more readily. The effect of crowding is then analysed in 3 different reaction systems - linear (representing metabolic pathways), parallel (ribosomal) and a combined GBA (growth balance analysis) model (representing balanced cell growth), previously introduced by one of the authors.

Through simulations the authors find that the optimal cytosolic occupancy (wrt both maximisation of reaction fluxes and growth rate) is lower for ribosomal reaction systems than for metabolic reaction systems, suggesting a trade-off between facilitating fast metabolism vs fast protein biosynthesis. In their combined model they further find that for various environmental conditions (parameterised by external nutrient concentration and number of enzymes required) the optimal cytosolic occupancy is consistent with experimental estimates of occupancy in corresponding growth conditions.

I have a concern regarding the derivation of the effective Michaelis parameter and some suggestions regarding presentation:

1) In the first part of the methods, the crowding-adjusted Michaelis parameter K_m^* is derived, however, the derivation is based on K_m ~ k_off/k_on (and later equality is used) rather than the conventional definition of K_m = (k_off+k_cat)/k_on. The change of definition is not commented on but should have effects on the overall derivation. Please can the authors comment on this choice of definition and its impact on the results?

2) I wonder to what extent the results could be strengthened via an analytical derivation of the optimal cytosolic occupancy. There is mention of an iterative procedure due to the interdependency of [ES] and K_m^*, but the algorithmic details are not specified and it is unclear to me whether at least for the linear and parallel models analytical results may be feasible.

3) After [Disp-formula pcbi.1011177.e005] it is mentioned that transition-state limit dominates when \\theta -> \\inf, but it is not obvious to see in [Disp-formula pcbi.1011177.e005] how this would result in K_M^*=K_M^0. After more careful reading of the methods section I understood that in the ideal ground case also \\Gamma=1, which then gives equality. This could be presented more clearly.

3) Figures could be presented more concisely. For example, there seems to be redundancy between Fig 5 (which also doesn’t have an explanatory caption) and Fig S2. In Fig 7, panels are not labelled, and the figure also looks more appropriate for the SI (to support Fig 6). Overall, the content of figures is presented very generously regarding space on the page, the authors may want to consider smaller plots and perhaps using insets (e.g. combining Figs 4 & 5).

4) I feel an overview schematic, illustrating the effects of the parameters \\Gamma, \\rho and \\theta, could improve clarity of presentation.

5) I would suggest a consistent use of math fonts to clearly indicate variables in the main text.

Overall, I think this is a study that could be of interest to the readership of the journal once the concerns (particularly 1)) have been addressed.

**Have the authors made all data and (if applicable) computational code underlying the findings in their manuscript fully available?**

Reviewer #1: Yes

Reviewer #2: Yes

PLOS authors have the option to publish the peer review history of their article (what does this mean?). If published, this will include your full peer review and any attached files.

Reviewer #1: No

Reviewer #2: No
---

## [Decision Letter · Decision Letter 1]

11 Apr 2023

Dear Prof Lercher,

Thank you very much for submitting your manuscript "Optimal density of bacterial cells" for consideration at PLOS Computational Biology. As with all papers reviewed by the journal, your manuscript was reviewed by members of the editorial board and by several independent reviewers. The reviewers appreciated the attention to an important topic. Based on the reviews, we are likely to accept this manuscript for publication, providing that you modify the manuscript according to the review recommendations.

Sincerely,

Stefan Klumpp

Academic Editor

PLOS Computational Biology

James O'Dwyer

Section Editor

PLOS Computational Biology

Reviewer's Responses to Questions

**Comments to the Authors:**

Reviewer #1: The authors have responded to most of the reviewers' criticism. Discussions on other possible mechanisms and their relation to other literature are included in the updated manuscript. Most importantly, the authors performed a more thorough analysis in comparison with the experimental data from ref. 10, and the claim is toned down that the current analysis is consistent with those data, but not necessarily the only explanation of the observation.

With the new analysis regarding data with ref. 10, I still have a concern as below and I strongly hope that the authors add a discussion on this. If that is done in a convincing manner, I believe this paper is worth publishing to raise the discussion of this interesting claim in the community.

What I am concerned about by looking at the newly added statistical analysis in Table S1A is that it shows the \\rho_DM for MM+gly medium is significantly different from any other growth condition and the authors ignore it in the analysis in the main text. In fact, I agree intuitively with the authors that the data feels that only RDM is truly different (they also have extremely low p-value). But if we accept only the extremely low p-value only, then I would say Table S1B is showing that only RDM is different for \\rho and not other conditions. Of course, one should not tune the p-value threshold after performing the analysis. Just it feels somewhat strange that these features are ignored in the main text.

In addition, my feeling is that the real source of these somewhat inconclusive analyses is the fact that the data have a large variance. This gives me another concern that even if the median may be significantly different for \\rho or \\rho_DM, the real cells do not seem to be really optimized to that value - in other words, the cells are perfectly fine to deviate from the value of \\rho individually from the predicted optimum. If there is selection pressure to converge to the predicted optimum, shouldn’t that also select for reducing the variance, too? How much benefit in the growth rate does the population get, if there is such a wide deviation?

Overall, discussing the statistical significance of the median does not solve my concern in terms of the experimental evidence. It will be desirable if the authors can thoroughly discuss how the authors' hypothesis explains the variance of the data.

Reviewer #2: My concerns have been satisfactorily addressed.

**Have the authors made all data and (if applicable) computational code underlying the findings in their manuscript fully available?**

Reviewer #1: Yes

Reviewer #2: Yes

PLOS authors have the option to publish the peer review history of their article (what does this mean?). If published, this will include your full peer review and any attached files.

Reviewer #1: No

Reviewer #2: No

Figure Files:

Data Requirements:

Reproducibility:

References:

---

## [Editor Report · Decision Letter 2]

11 May 2023

Dear Prof Lercher,

We are pleased to inform you that your manuscript 'Optimal density of bacterial cells' has been provisionally accepted for publication in PLOS Computational Biology.

Best regards,

Stefan Klumpp

Academic Editor

PLOS Computational Biology

James O'Dwyer

Section Editor

PLOS Computational Biology

---

## [Editor Report · Acceptance letter]

1 Jun 2023

PCOMPBIOL-D-22-01596R2 

Optimal density of bacterial cells

Dear Dr Lercher,

I am pleased to inform you that your manuscript has been formally accepted for publication in PLOS Computational Biology. Your manuscript is now with our production department and you will be notified of the publication date in due course.

With kind regards,

Zsofi Zombor
